# DittoGym: Learning to Control Soft Shape-Shifting Robots

**Suning Huang**[†]
Department of Automation
Tsinghua University
`hsn19@mails.tsinghua.edu.cn`

**Boyuan Chen**
CSAIL
Massachusetts Institute of Technology
`boyuanc@mit.edu`

**Huazhe Xu**
IIIS
Tsinghua University
`huazhe_xu@mail.tsinghua.edu.cn`

**Vincent Sitzmann**
CSAIL
Massachusetts Institute of Technology
`sitzmann@mit.edu`

## Abstract

Robot co-design, where the morphology of a robot is optimized jointly with a learned policy to solve a specific task, is an emerging area of research. It holds particular promise for soft robots, which are amenable to novel manufacturing techniques that can realize learned morphologies and actuators. Inspired by nature and recent novel robot designs, we propose to go a step further and explore the novel *reconfigurable* robots, defined as robots that can change their morphology within their lifetime. We formalize control of reconfigurable soft robots as a high-dimensional reinforcement learning (RL) problem. We unify morphology change, locomotion, and environment interaction in the same action space, and introduce an appropriate, coarse-to-fine curriculum that enables us to discover policies that accomplish fine-grained control of the resulting robots. We also introduce Ditto-Gym, a comprehensive RL benchmark for reconfigurable soft robots that require fine-grained morphology changes to accomplish the tasks. Finally, we evaluate our proposed coarse-to-fine algorithm on DittoGym and demonstrate robots that learn to change their morphology several times within a sequence, uniquely enabled by our RL algorithm. More results are available at https://dittogym.github.io.

## 1 Introduction

Over millions of years, morphologies of species change as a function of evolutionary pressures (Minelli, 2003; Raff, 2012). In robotics, this process of evolution has inspired the task of robot co-design: the joint optimization of a robot's morphology and a control policy that best enable the robot to accomplish a given task (Gupta et al., 2022; Wang et al.; Ha, 2019; Yuan et al., 2021).

Yet, in nature, creatures do not only change their morphology over millions of years as a function of evolution. Almost all living beings go through a process of morphology changes even in their lifetime. These changes can be dramatic in magnitude, like when a mighty tree grows from a tiny sapling, but they can also be dramatic in *form*, like across the many examples of metamorphosis, where frogs, for instance, go through a water-dwelling stage with a tail for propulsion, to then lose their tail and grow legs to live on land (Rose, 2005; Hofmann et al., 2003).

Inspired by this process, we propose the study of *reconfigurable* soft robots. Reconfigurable soft robots, as the name suggests, are robots that are capable of dynamically changing their shape and morphology during their lifetime to accomplish varying tasks. Resembling organisms in nature that change their morphology in response to evolving conditions (Ning et al., 2023), reconfigurable soft robots offer an exciting avenue for addressing complex real-world challenges. While the actual manufacture of reconfigurable soft robots will remain a challenging problem for some time, recent

---

[†]Work done as a visiting researcher at MIT.
  Code is available at DittoGym and CFP.

advances have seen the first implementations, such as a recent design manufactored from a ferro-magnetic slime (Sun et al., 2022) that has exciting applications in healthcare.

In this paper, we identify three key challenges in the algorithmic study of reconfigurable robots and propose means to overcome them.

Firstly, while it is easy to describe what a reconfigurable robot *is*, there is no consensus about how to simulate these robots, and how one may parameterize their actions. In this paper, we propose to simulate soft reconfigurable robots via the Material Point Method (MPM) while modifying the Cauchy stress update to incorporate actuation specified via a continuous muscle field. This leads to a simulation that models the recently proposed reconfigurable magnetic slime robot (Sun et al., 2022). As a result, we are able to formalize reconfigurable robot control into a reinforcement learning problem with a 2d continuous muscle field as its action space.

Secondly, the unstructured nature of soft robots renders traditional control algorithms unsuitable (Lee et al., 2017). While learning based methods like reinforcement learning excel in a variety of unstructured control problems (Li, 2017), reconfigurable robots pose a unique challenge due to the extremely high-dimensional action space necessary for fine-grained morphology change. For instance, for RL to succeed, we need to ensure that random exploration will lead to meaningful morphology changes (Tang et al., 2017). In practice, this means that actions have to move large chunks of the robot at a time. On the flip side, to execute fine-grained actions such as walking, the action space needs to support fine-grained actuation. As we will see, trying to learn a fine-grained policy via existing reinforcement learning fails. We hence design a multi-scale muscle field, where a policy with coarse discretization for actions is trained first to actuate large chunks of the robot to effect meaningful morphology changes while a fine-grained policy is learned on top. We benchmark our algorithm with strong baselines and find that our coarse-to-fine framework is the only algorithm that can control the reconfigurable robot to accomplish multi-stage tasks in our benchmark.

Finally, there is no standard benchmark for fine-grained reconfigurable robots. We introduce *Ditto-Gym*, the first RL benchmark for reconfigurable robots. DittoGym is a suite of eight long-horizon tasks that require varying degrees of morphological change. Four environments require the robot to change its morphology *more than once* during the execution of the task.

In summary, our contributions are three-fold. Firstly, we formalize the joint control of morphology and policy of deformable soft robots into a unified reinforcement learning problem under a continuous 2D muscle field. Secondly, we propose **C**oarse-to-**F**ine **P**olicy (CFP), a novel RL algorithm with a hierarchical action space that enables such robots to explore meaningful actions to control a deformable soft robot. Thirdly, we build a comprehensive benchmark, DittoGym, that evaluates control algorithms for deformable soft robots in complex tasks requiring dynamical morphological changes.

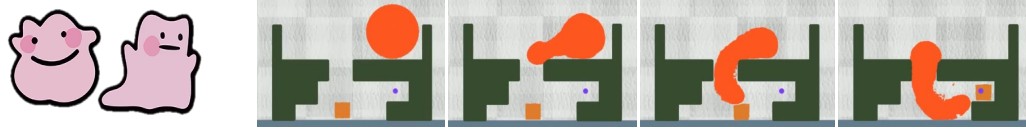

Figure 1: **Reconfigurable soft robots.** In this paper, we address challenges in controlling reconfig-urable - or shape-shifting - robots, who can change their morphology to accomplish desired tasks. The figure above illustrates a task where a circular robot needs to alter its body shape to fit within a confined chamber to manipulate the square cargo to the target point (right). We introduce a bench-mark with 8 tasks for shape-shifting robots, which we dub "DittoGym" inspired by a shape-shifting Pokémon (left).

## 2    RELATED WORK

**Robot design simulators.**    The field of automated robot morphology design has garnered significant attention, resulting in a variety of simulators suitable for testing and simulation. Broadly, these simulators can be classified into two categories: those designed for rigid robots and those for soft robots. For simulating rigid robots, representative simulators include Mujoco (Todorov et al., 2012)

and RoboGrammar (Zhao et al., 2020). In these simulators, the basic components of the robot are predefined, and the robot's morphology is parameterized using a graph-like structure. While these simulators offer a straightforward representation of morphology, they lack flexibility and generalizability in design and are limited to simulations of rigid robots. In contrast, for simulating soft-body robots, representative simulators include EvoGym (Bhatia et al., 2021) and Softzoo (Wang et al., 2023a). These simulators focus on the co-design of soft-body robot morphology and control. However, prior work using these simulators exclusively focuses on *co-design*, with no prior work investigating reconfigurable soft robots. Consequently, these simulators primarily focus on fully elastic soft-body robots, not accounting for robots that can undergo plastic deformation to accomplish a task. A notable gap exists in the lack of a simulator capable of accurately simulating highly reconfigurable soft robots. Therefore, we have developed a novel simulation platform, DittoGym, designed to accurately simulate reconfigurable robots for the study of their motion and control. The parameters of our simulation platform are adjustable, enabling it to simulate a range of robot types, from near-rigid robots to completely elastic soft-body robots.

**Robot design algorithms.** Many co-design algorithms adopt a common strategy of exploring the morphology design space. For instance, RoboGrammar and SoftZoo (Zhao et al., 2020; Wang et al., 2023a) employ a heuristic search algorithm based on morphology grammar, while previous studies have utilized evolutionary algorithms to optimize robot morphology by maintaining numerous morphology candidates (Cheney et al., 2014; Gupta et al., 2021). However, these search-based zeroth-order optimization methods are computationally demanding and inefficient (Golovin et al., 2019). An alternative research direction aims to seamlessly integrate robot morphology into robot states and learn end-to-end control policies (Yuan et al., 2021). Recent efforts have also tried to learn latent space of morphologies using latent-variable models such as Variational Autoencoders (Hu et al., 2023; Ha et al., 2021; Wu et al., 2020) and Diffusion model (Wang et al., 2023b), though they either rely on predefined graph grammars or need pre-training on a large dataset. Despite the various branches that have emerged from co-design methods, they all result in static morphologies that cannot be modified during the robot lifetime, making them unsuitable for reconfigurable robot motion and control.

## 3 PRELIMINARY

**Material Point Method.** The Material Point Method (MPM) is a versatile computational particle-based simulation method widely employed in the fields of solid mechanics and computational physics (Jiang et al., 2016; Hu et al., 2018; Han et al., 2019; Ram et al., 2015; Stomakhin et al., 2013; Russell & Celia, 2002). As a hybrid of Eulerian and Lagrangian methods, MPM involves both grid-based and particle-based operations. The grid stage primarily serves to handle boundary conditions, while the particle stage is focused on addressing interactions among thousands of particles. In this latter stage, a critical task is to compute the material strain, also known as Cauchy stress, as it profoundly influences the subsequent motion of particles. The Cauchy stress is determined by the following equation: $Cauchy\ Stress = 2\mu(\mathbf{F}_p - \mathbf{r})\mathbf{F}_p^T + \text{diag}(\lambda(\mathbf{J}_p - 1)\mathbf{J}_p)$ Here, $\mu$ and $\lambda$ represent material parameters, $\mathbf{r}_p$ is an orthogonal matrix obtained from the Polar Decomposition in MPM, and $\mathbf{J}_p$ is the determinant of $\mathbf{F}_p$.

**Von Mises yield criterion.** The von Mises yield criterion is employed to predict when a material will undergo plastic deformation (Madenci & Oterkus, 2016). The deformation gradient $\mathbf{F}_p$ can be decomposed using Singular Value Decomposition (SVD) to obtain a diagonal matrix $\mathbf{\Sigma}_p$, which represents the deformation scale for each particle. Plastic deformation initiates when the norm of this matrix, denoted as $\|\mathbf{\Sigma}_p\|_2$, exceeds the yield criterion $Yield_m$. At this stage, the material "forgets" its initial state, necessitating a deformation gradient projection to account for this deviation from the original configuration.

**Reinforcement learning.** We employ Reinforcement Learning (RL) to train the control policy for our highly reconfigurable robot. In the realm of RL, we model the problem as a Markov Decision Process (MDP), characterized by the quintuple $(S, A, P, r, \gamma)$. Here, $S$ signifies the state space, $A$ denotes the action space, $P : S \times A \to S$ encapsulates the environment's transition dynamics, $r(s, a) : S \times A \to \mathbb{R}$ quantifies rewards, and $\gamma \in (0, 1]$ governs the temporal discounting of rewards.

Our RL agent's objective is to derive a policy $\pi_\theta(a_t|s_t)$, parameterized by a deep neural network, aimed at maximizing the expected cumulative reward $E_\pi \left[ \sum_{t=0}^\infty \gamma^t r(s_t, a_t) \right]$.

## 4 METHOD

In this section, we aim to address the three aforementioned challenges in the algorithmic research of reconfigurable soft robots, namely lack of formalization, lack of appropriate algorithm and the lack of a benchmark.

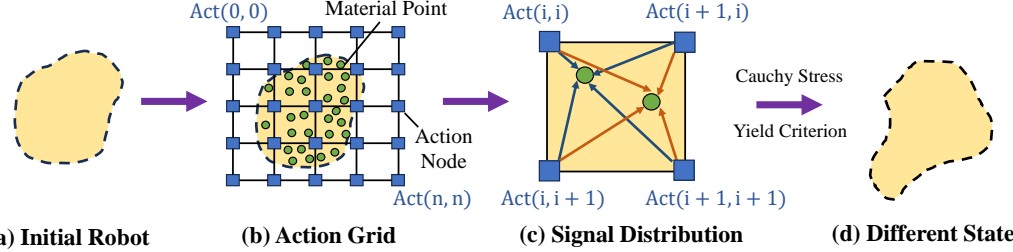

Figure 2: (a) The soft reconfigurable robot, initialized with a specific shape, is ready to venture into the task. (b) Instead of directly controlling material points in the robot, the policy first applies actions to the action grid, which corresponds to the grid in the material point method (MPM). (c) Each particle aggregates actuation signals from its adjacent grid points during the grid-to-particle distribution stage. (d) Under the principles of Cauchy stress and the von Mises yield criterion, we can actuate the particles in a physically plausible manner, thereby changing the robot's morphology and state.

### 4.1 FORMALIZATION

Controlling reconfigurable robots is an unstructured control problem that is naturally suited for trial-and-error methods such as reinforcement learning. We thus formalize the control problem into building blocks of an MDP as defined in Section 3.

In order to define the transition dynamics in a MDP, we need to formalize reconfigurable soft robots into a simulatable system while staying close to real hardware.

To do so, we adopt the MPM as the simulation backbone for soft robots. As shown in in Figure 2, to simulate realistic deformations, we draw inspiration from prior work (Huang et al., 2021; Gao et al., 2017), introducing the von Mises yield criterion into the MPM algorithm. This criterion models the robot as elasto-plastic materials, exhibiting elastic deformation at low strains and plastic deformation at higher strains. This physical property is exhibited by many real-world materials, including the material most recently used to build a real-world reconfigurable robot (Sun et al., 2022). We also introduce a modification to the Cauchy stress equation (Rothenburg & Selvadurai, 1981; Hu et al., 2019b), allowing particles to stretch or compress their shape in response to an action signal, analogous to natural muscles. Specifically, this additional term can be expressed as a multiplication of the gradient deformation matrix and action signals: $\mathbf{F}_p \Sigma_p^c \mathbf{F}_p^T$, where $\Sigma_p^c$ is a second-order diagonal matrix designed to receive a 2D action signal vector. This operation directly alters the strain of the particle along its two principal axes. Consequently, the final Cauchy stress in an MPM simulation can be expressed shown in Equation 1. This modification enables us to explicitly assign a 2-dimensional action signal vector $\mathbf{A}_p$ to each particle's Cauchy diagonal matrix ($\mathbf{\Sigma_p^c}$), governed by deformation equations known in material science. As a result, the robot exhibits macroscopic changes in morphology when an action forces a significant portion of the particles to undergo plastic deformation, while stress below the von Mises yield stress leads to elastic deformations, serving as a spring-like mechanism to provide, for instance, means for locomotion.

$$\textit{Final Cauchy Stress} = 2\mu(\mathbf{F}_p - \mathbf{r})\mathbf{F}_p^T + \text{diag}(\lambda(\mathbf{J}_p - 1)\mathbf{J}_p) + \mathbf{F}_p \Sigma_p^c \mathbf{F}_p^T \qquad (1)$$

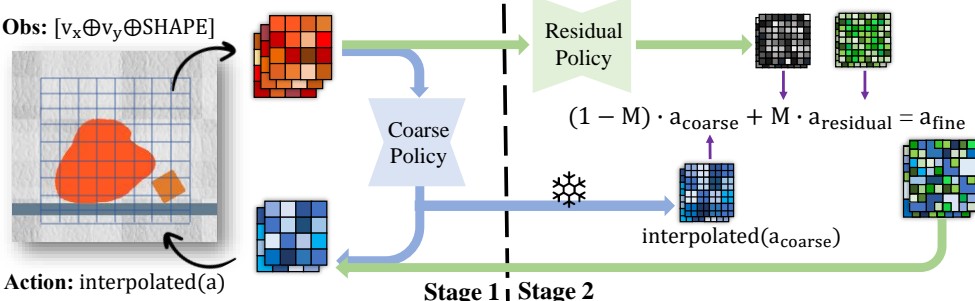

Figure 3: **Illustration of CFP.** In stage one, we train a coarse policy to efficiently explore the action space and discover meaningful action patterns. Subsequently, in stage two, we employ a coarse-to-fine approach to train a high-resolution residual policy that delves deeper into optimizing the actions for improved performance.

The above MDP transition dynamics require us to specify an actuation vector for every single particle. However, we note that MPM is a particle-based simulation method that approximates continuous dynamics in continuous space. Hence, action space should similarly be a continuous field where each coordinate within the robot has an action vector. This also mirrors the control of the magnetic slime robot via an external magnetic field (Sun et al., 2022). We illustrate this in Figure 2.

With the action space defined, it is easy to define the remaining components of our MDP: The state is simply the state of all particles in the MPM simulation, whereas we can choose observation and reward functions in a case-by-case manner for different environments.

We have thus formalized the abstract problem of controlling a reconfigurable robot into an MDP problem that can be solved via reinforcement learning.

## 4.2 ALGORITHM

The model is designed with a fully-convolutional architecture, enabling it to analyze pixel-level observations of the robot's shape and its velocity. Based on this analysis, it generates discrete grid action signals, which are instrumental in approximating the ideal force field, as shown in Figure 3. Please refer to Appendix D for more detailed illustration of CFP.

**Discretizing the action parameterization.** Our MDP requires an infinite-dimensional action space in the form of a continuous 2D force field acting on the robot. To parameterize this continuous force field, we note that bicubic interpolation can interpolate a regular n-D grid of features into a mapping from any n-D coordinate to a feature. We thus parameterize the policy via a neural network that outputs a discrete n-D grid of actions and interpolate it onto our continuous action space.

**Fully-convolutional policy network.** While the discretized parameterization provides a differentiable way to specify a spatially continuous action, we require a discretization resolution that is high enough to control fine-grained morphology changes. High resolutions lead to a high-dimensional action space, which has proven to be challenging for traditional reinforcement learning algorithms (Duan et al., 2016; Metz et al., 2017). However, we note that in our two-dimensional simulator, observations are 2D images that contain both the robot as well as its immediate environment (see Figure 3). Further, actions act on the same 2D neighborhoods that we observe in the image. We thus propose to utilize a fully convolutional autoencoder policy network to parameterize the policy, which can share parameters across spatial locations, effectively reducing the size of the search space. This network takes images of the reconfigurable robot and its immediate environment as input, and generates a 2D action output that approximates the continuous muscle field via bicubic interpolation. Additionally, drawing inspiration from (Jiang et al., 2015), we adopt a grid-to-particle approach to smoothly distribute the grid output to individual particles.

**Coarse-to-fine policy learning.** As highlighted in Section 1, random actions at high resolution will tug and pull at single particles without leading to a significant macroscopic morphology change, leading to meaningless parameter updates to the policy. To address this issue, we observe that action

output at a coarser resolution, when interpolated, actuates large muscle chunks to move together due to the increased grid size. Such coarse discretization often deforms the soft robot substantially even under randomly sampled actions. Therefore, we first train a coarse policy, denoted as $\pi_{\theta_{\text{coarse}}}$, within our Markov Decision Process (MDP) framework, leveraging low-resolution action discretization for expedited exploration of significant morphology changes.

Nevertheless, relying solely on the coarse policy proves insufficient for achieving success in complex tasks, as it does not harness the complete control granularity available within the action space. To overcome this limitation, we introduce a refinement stage: we freeze the parameters of the trained coarse policy $\pi_{\theta_{\text{coarse}}}$ while training a new residual policy with increased resolution atop the predicted coarse actions. Specifically, we concatenate the observation with the coarse policy's output as input to the fine policy neural network. The fine policy then produces a residual action prediction $\mathbf{a}_{\text{residual}}$ and a gating mask $\mathbf{M}$ with values between $(0, 1)$ via sigmoid activation. Utilizing this mask, we blend the coarse policy and residual policy via a weighted sum:

$$\mathbf{a}_{\text{fine}} = \mathbf{M} \cdot \mathbf{a}_{\text{residual}} + (1 - \mathbf{M}) \cdot \mathbf{a}_{\text{coarse}}$$

It is important to note that in this equation, the output from the residual policy is of higher resolution compared to the coarse action, necessitating upsampling of coarse actions before applying the mask.

The utilization of a gated mask, while freezing the fixed $\pi_{\theta_{\text{coarse}}}$, renders $\pi_{\theta_{\text{fine}}}$ less susceptible to local optima during exploration of the high-dimensional action space. Simultaneously, the gated mask selectively overrides coarse policy predictions when finer actions are deemed more appropriate, allowing for much finer control granularity to maximize expected returns across various tasks.

This coarse-to-fine training methodology can be iteratively applied to incrementally scale up the resolution of the predicted action. In our implementation, we set the output resolution of the residual policy to be twice that of the coarse policy output.

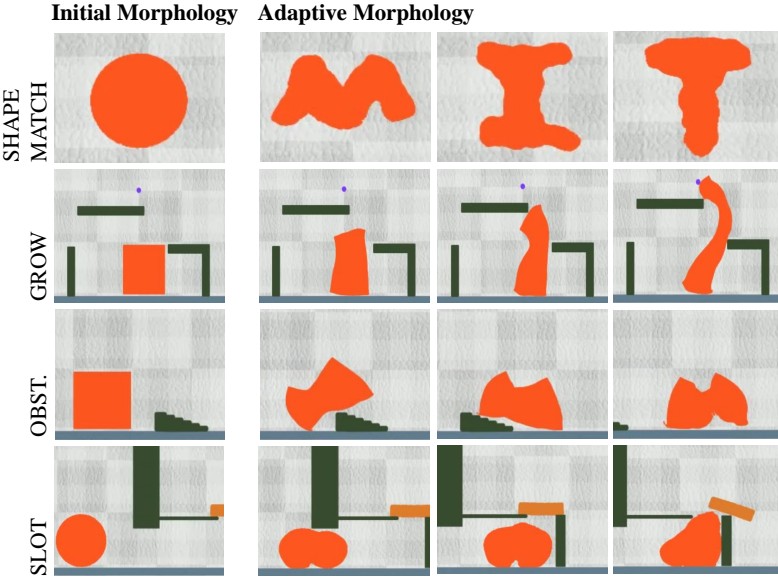

Figure 4: **Illustration of reconfigurable robots in diverse tasks.** This figure showcases selected visualization results from DittoGym. Notably, policies trained under the guidance of CFP exhibit precise control over highly reconfigurable robots, enabling them to successfully accomplish their respective tasks. More visualization results can be found in Appendix C.

## 4.3 BENCHMARK

Digital benchmark environments are widely utilized by control algorithms researchers to expedite their research and facilitate fair comparisons (Yu et al., 2020; Bhatia et al., 2021). Our benchmark, DittoGym, addresses the absence of standardized benchmarks for reconfigurable soft robots with fine-grained morphology changes. We incorporate our formalized simulation (Section 4.1) into a set

of OpenAI Gym environments. To model deformable soft robots and external objects for interaction, we implement the simulation of relevant materials in the Material Point Method (MPM) using the Taichi (Hu et al., 2019a) framework. Consequently, DittoGym not only provides a user-friendly Python interface but also fully leverages hardware accelerators, such as GPUs, to simulate MPM particles. We choose the observation space to be a rasterized 2D image of the square area around the geometric center of the robot, with both occupancy and velocity as channels. The action space is implemented as a 2D actuation grid for the same square with the highest possible resolution for simulation (MPM grid resolution). As mentioned in Section 3, each 2D vector in the 2D action grid controls the stress applied to the corresponding point, where the magnitude determines whether plastic or elastic deformation occurs.

Under this framework, we implemented a variety of tasks covering shape matching, locomotion, reaching, and manipulation. Each task necessitates morphological adaptation to meet specific fine-grained requirements, sometimes requiring multiple morphological changes to achieve the desired goals. We provide a brief overview of these eight tasks in Appendix A.

## 5 EXPERIMENTS

In this section, we hope to answer the following questions via empirical evaluation. 1) Can our formalization, when combined with appropriate algorithms, enable simulated soft robots to perform interesting tasks that require fine-grained morphology changes? 2) Can our benchmark DittoGym sufficiently evaluate algorithms designed to control *fine-grained* morphology changes for soft robots? 3) Compared to relevant baselines, how sample efficient is CFP for reconfigurable soft robot control?

### 5.1 INTERESTING TASKS WE ENABLE

One of our main contributions is the formalization of reconfigurable robot control into a reinforcement learning problem. In this section, we demonstrate a series of tasks a simulated reconfigurable robot can uniquely perform under this formalization when combined with appropriate algorithms. Figure 1 illustrates an interesting multi-stage task made possible with our framework. In Figure 4, we show four additional tasks from DittoGym that require the robot to alter its morphology during execution.

In the SHAPE MATCH task, the robot is able to successfully mimic the target alphabet shape, highlighting the robot's ability to reconfigure to arbitrary target shapes. In the GROW task, the robot learns to elongate and curve its body. This novel shape enables the robot to weave its way around obstacles to reach the target point, similar to a seed germinating. In the OBSTA-CLE task, the robot elongates its body and leans forward in order to leverage the force of gravity to swiftly stride over the obstacle. Subsequently, the robot grows two leg-like stumps and uses them to walk, enabling efficient for-

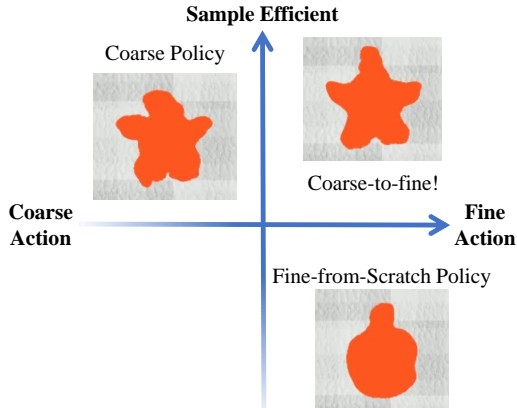

Figure 5: **Qualitative analysis of different methods.** A policy learns to shape the robot into a star. Fine-from-scratch policy suffers from exploration as random actions at high resolution will tug and pull at single particles, preventing meaningful morphology changes. A coarse policy have higher sample efficiency as it can make meaningful explorations, but cannot take advantage of fine-grained action space. CFP takes advantage of both.

ward progress. Lastly, for the SLOT task, the robot strategically reduces its height while growing two very short legs to maneuver through an exceptionally narrow space, resembling a mouse running through a pipe. To open the wooden lid at the end of the pipe, the robot ungrows its legs and grows a tall torso to manipulate the lid effectively.

## 5.2 Whether DittoGym necessitates fine-grained action

Before we can quantitatively evaluate CFP against baselines in DittoGym, it's necessary to prove that DittoGym actually requires *fine-grained* morphology changes for the soft robot to achieve high rewards.

Quantitatively, we take the expert policies trained under each action space granularity and visualize the maximum reward they can achieve. We study this metric for three different levels of granularity, coarse ($4 \times 4 \times 2$), medium ($8 \times 8 \times 2$) and fine ($16 \times 16 \times 2$). As illustrated in Figure 9, higher action resolution almost always leads to higher possible reward across all 8 environments. This trend confirms that DittoGym requires the agent to learn fine-grained morphology changes.

Qualitatively, we can visualize the expert policies' behaviors under different levels of granularity. In Figure 5, we visualize the best shape the robot can achieve in the SHAPE MATCH task under different action resolutions. Under the coarser resolution, the robot can make meaningful changes towards the target shape but fails to generate high-frequency detail of the target shape, such as the tips of each arm of the star. On the other hand, the high-resolution action parameterization allows the robot to match the star shape well.

## 5.3 Sample efficiency of CFP

Finally, we hope to understand whether CFP has higher performance compared to baselines when controlling reconfigurable robots with fine-grained morphology changes. We therefore benchmark CFP and aforementioned baselines in 8 different environments in DittoGym under 3 seeds.

**Baselines.** We benchmark our coarse-to-fine, fully convolutional action parameterization with prominent alternatives:

- **GNN Policy:** Modular-based control emerges as a potential solution for our challenge, as it enables the treatment of a robot as distinct modules and offers the flexibility to adapt to various robot topologies. However, these methods face limitations in direct application to our context, as highly-reconfigurable robots often lack articulated joints and links, and therefore do not possess the explicit topological structure required by these baseline methods. In spite of these challenges, we integrate the well-established Modular Policy baseline (Pathak et al., 2019) into our problem framework. To effectively modularize the robot, we utilize k-means clustering to assign topology, thereby allowing the application of the Graph Neural Network (GNN) from the original method. At each step, the robot is initially segmented using the unsupervised K-means method. Following this, we employ the state information of each mass center point, treated as joints, along with the adjacency relationships of the segments, to construct the graph. This approach enables us to harness a standard RL framework for training the control policy.

- **Transformer Policy:** The GNN-based modular policy requires an explicit assignment of adjacency relationships among all segments, potentially limiting information propagation during training. To address this, we introduce an additional baseline wherein an attention mechanism autonomously determines the topology for information processing. A more comprehensive explanation and visualization of modular-based approaches is available in Appendix E.

- **Coarse Policy (ablation):** This baseline only trains a coarse policy ($8 \times 8 \times 2$) but does not implement the coarse-to-fine technique.

- **Fine-from-Scratch Policy (ablation):** This baseline directly trains a fine policy ($16 \times 16 \times 2$) without implementing the proposed coarse-to-fine curriculum. The intention of this baseline is to investigate whether, under the current framework, an agent can learn an efficient high-dimensional policy without the coarse-to-fine process.

In Figure 6, we plot the episode reward curves for all the runs. Our proposed algorithm, CFP, outperforms all baselines consistently across all tasks in terms of overall sample efficiency. In addition, CFP can consistently attain higher episode reward upon convergence. When we visualize the policies, only CFP is able to succeed in long-horizon multi-stage environments, presenting it as the only viable option to solve these complex tasks.

Although in CATCH task, the best Transformer-based baseline could achieve same-level performance as CFP, it is highly oscillated and unstable. In most tasks, the baselines does not perform well. For

each GNN and Transformer-based method, we implement a sparse-segment version (with 16 segments) and a dense-segment version (with 64 segments). It's not unexpected that the modular-based control policy, originally developed for addressing control issues in robots with clearly defined topological structures, falls short when applied to this innovative reconfigurable robot. More comparison results can be found in Appendix E.

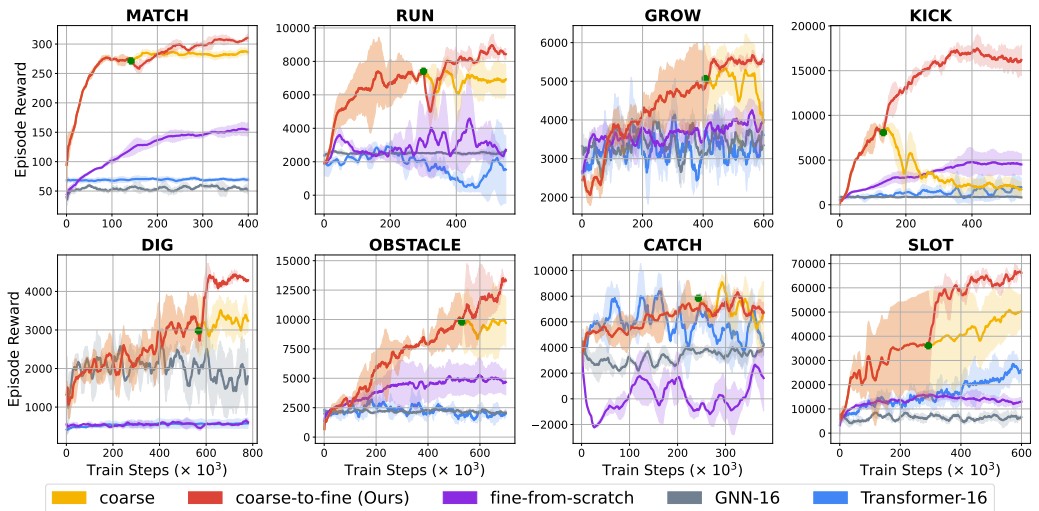

Figure 6: **Comparison of fine-resolution policies trained with and without coarse-to-fine technique.** The plots showcase the underperformance of a fine-resolution policy trained from scratch compared with a coarse-to-fine result. It further reflects that parametrizing the action space via modular policies (GNN-based and Transformer-based) still fall short of the coarse-to-fine technique. The green dot represent the checkpoint where we start to implement coarse-to-fine technique.

Why is coarse-to-fine so good? In Figure 5, qualitative visualization of different methods shows that fine-from-scratch fails to match the shape of the star despite having a sufficient control resolution to do so. This can be explained by the exploration problem brought by high-dimensional action spaces as we detailed in Section 4.2. On the other hand, while the coarse policy isn't fine-grained enough to get a perfect star shape, it has a lower-dimensional action space which actuates larger chunks of muscles together, leading to meaningful morphology explorations. However, the performance of coarse policy is ultimately limited by its granularity and would produce a star shape that's far worse than the one produced by our coarse-to-fine method.

Both the quantitative comparison and qualitative analysis demonstrates the effectiveness of CFP in DittoGym, illustrating its unique ability to control reconfigurable soft robots to accomplish long-horizon tasks that requires fine-grained shape changes.

# 6 CONCLUSION

Many creatures in nature change their morphology throughout their lifetime. Finding ways in which we may simulate and train reconfigurable robots that mimic this is impactful across robotic applications, with first practical examples already proposed Sun et al. (2022). This paper establishes a formal reinforcement learning framework for investigating and controlling reconfigurable soft robots. We first introduce an innovative parameterization of reconfigurable robots as a soft material actuated via a muscle field. We then present a framework, CFP, which leverages a fully-convolutional state estimation and action parameterization, enhanced by a coarse-to-fine curriculum that guarantees meaningful exploration while preserving fine-grained control capabilities. We introduce the first benchmark, DittoGym, with eight environments that require drastic morphology changes to accomplish the embedded tasks. We demonstrate that our robots, trained by CFP, learn to dynamically alter their morphology several times throughout their lifetime to accomplish these tasks. We significantly outperform baselines that do not employ a coarse-to-fine parameterization of the action space.

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

APPENDIX

# A    BENCHMARK DETAILS

## A.1    TASKS

The DittoGym includes eight distinct types of tasks encompassing shape matching, locomotion, reaching, and manipulation. The environment settings and task objectives for each of these tasks are detailed as follows:

- **SHAPE MATCH:** The robot is initialized as a circle in a zero-gravity environment. It has to alter its shape to mirror predefined geometric or alphabetical form. The score is calculated on the basis of the congruence between the robot's current shape and the target shape.
- **Locomotion - RUN:** The robot is initialized on a plain and the task requires a robot to move forward to the greatest possible distance within a stipulated period. The score is governed by the distance traversed and the speed maintained.
- **Locomotion - KICK:** The robot is initialized as a circle and is designed to kick a square target to the maximum possible distance. Ground friction prevents simple pushing, therefore the robot is required to use a flipping and rolling technique.
- **Locomotion - DIG:** The robot is initially in the shape of a circle placed on the top of a soil-filled container, aiming to reach a target located at the bottom-right corner.
- **Locomotion - OBSTACLE:** The robot, in the shape of a square, faces an obstacle in its path while the task is to move forward. The score is based solely on how far the obstacle is bypassed.
- **Reaching - GROW:** The robot is initialized as a square on the ground and is required to extend its superior segment to reach a target represented by a purple dot. Multiple obstacles hinder the direct route to the target. The reward function measures the distance between the target point and the robot.
- **Manipulation - CATCH:** The circular robot, placed outside a cube, is tasked with manipulating a small square target within the cube to a specific point. The score is computed considering the distance between the robot and the cube, as well as the cube and the final point.
- **Manipulation - SLOT:** The circular robot, initially outside a box with only a narrow slot to get inside, must squeeze its body through the slot and manipulate a cap target on top of the box. The reward function measures the distance between the robot and the cap, and whether the cap was successfully removed from the box.

Our task selection follows both generic RL benchmark design principles and the specific needs of reconfigurable robots. We highlight the factors we hoped to cover in our benchmark and the corresponding factor vector of each environment as shown in Table 1:

- Tasks have to cover a range of difficulties (A=easy, B=hard).
- Some tasks evaluate robot itself, others require the robot's ability to interact with external objects (A=not require, B=require).
- Some tasks require just one morphology change, while others need changes of morphology multiple times (A=one change, B=multiple changes).
- Some have non-convex reward landscape that requires the algorithm to be long-horizon (A=short-horizon, B=long-horizon).
- Some features softer material and some features less soft material (A=softer, B=less soft).

## A.2    MPM PARAMETERS

Since this algorithm involves grid operations and requires a fixed grid defined in space as the computational domain for MPM, it is essential to define a relatively small grid space to enhance the computational efficiency of the MPM algorithm. However, in some locomotion tasks, highly reconfigurable robots may have movement distances that exceed the predefined maximum length of the grid area. To address this issue, we have adjusted the MPM algorithm by modifying the grid to

Table 1: **Benchmark Factors for Reconfigurable Robots.**

| Task | Factor Vector |
|------|---------------|
| SHAPE MATCH | [A, A, A, A, A] |
| RUN | [A, A, A, A, B] |
| GROW | [A, A, A, A, A] |
| KICK | [B, B, A, A, B] |
| DIG | [B, B, A, A, B] |
| OBSTACLE | [B, A, B, B, B] |
| CATCH | [B, B, B, B, A] |
| SLOT | [B, B, B, B, B] |

move along with the robot's center of mass during motion. This ensures that the robot always remains within the scope of the MPM algorithm, effectively resolving the challenge of efficiently simulating large-scale motions of the robot. The hyperparameters of MPM are shown in Table 2. Parameters in reference to the particle numbers, material characteristic and the maximum action depend on the tasks, which can be checked from our codes.

Table 2: **MPM Hyperparameters.**

| Name | Value |
|------|-------|
| p_vol | 1 |
| p_mass | 2 |
| n_grid | 128 |
| dt | 1e-4 |
| Repeat Times | 100 |
| Coeff | 0.5 |
| Bound | 1 |
| Seed | 0 |

# B TRAINING DETAILS

## B.1 MODEL STRUCTURE

In this project, we have adopted the SAC (Haarnoja et al., 2018) algorithm as the core framework for reinforcement learning. For the policy network architecture, we have implemented a fully-convolutional structure similar to the Nature CNN (Mnih et al., 2013). In addition to this, prior to each input, we have introduced a concatenation step where the image is combined with a position embedding channel. This augmentation significantly enhances the network's sensitivity to spatial positions. Regarding the critic network structure, we have employed an identical CNN encoder architecture as used in the policy network. Subsequently, we have integrated two layers of a 256-dimensional MLP network to compute and output the critic value. The code for this project will be publicly accessible upon acceptance. The network details are shown as follows:

**Critic Network.** The 3-layer CNN based encoder can embed the input of $64 \times 64 \times 5$ (3 channels for state, 2 channels for the upsampled action) images into a vector with 512 dims following a CNN architecture widely used in RL papers and frameworks (Raffin et al., 2019; Huang et al., 2021). Then it will get through a 3-layer MLP with 256 dims latent space to yield the Q value.

**Policy Network.** The similar 3-layer CNN based encoder can embed the input of $64 \times 64 \times 3$ images into a embedding shaped $4 \times 4 \times 32$ (still 512 dims). Then it will get through a 3-layer Deconvolutional network to yield grid-like action ($8 \times 8 \times 2$ or $16 \times 16 \times 2$). In the residual training stage, the policy network's input channel will also be 5 (3 channels for state, 2 channels for the upsampled coarse action).

## B.2    Reinforcement Learning

The specific parameters of the reinforcement learning algorithm we employed are presented in Table 3. If simulation alone, the environment can run at an average speed of 70.1 FPS on the RTX4090 GPU, with only 20-30% volatile utility usage and 15% memory usage. Training a single agent runs at approximately 20 FPS when executed on the same GPU. Parameters like alpha and max episode length depend on the tasks, which can be checked from our codes.

Table 3: **SAC Training Hyperparameters.**

| Name | Value |
|---|---|
| Batch Size | 256 |
| Auto Entropy-tuning | False |
| Gamma | 0.99 |
| Critic Network Hidden Size | 256 |
| Learning Rate | 3e-4 |
| Momentum | 0.99 |
| Optimizer | Adam |
| Replay Buffer Size | 200000 |
| Seed | 0,1 and 2 |

## C    Experiment Visualization

In Figure 7, we illustrate the eight tasks within the DittoGym: SHAPE MATCH, RUN, KICK, DIG, GROW, OBSTACLE, CATCH, SLOT.

In the SHAPE MATCH task, every robot is able to successfully mimic the target alphabet shape, highlighting the robot's ability to reconfigure to arbitrary target shapes.

In the RUN task, notable improvement is observed when the robot transforms into a feet-like structure. This adaptation aligns with the biological adjustments observed in many terrestrial organisms, underscoring the robot's ability to replicate natural processes effectively.

In the KICK task, the robot is required to exert force on an object to roll it on the ground, leading it to reshape into a configuration resembling an 'earth-mover', demonstrating an example of functional adaptation.

In the DIG task, the robot assumes a drill-like structure to meticulously navigate toward the target, showcasing a distinctive problem-solving approach and flexible adaptability.

In the GROW task, the robot learns to elongate and curve its body. This novel shape enables the robot to weave its way around obstacles to reach the target point, similar to a seed germinating.

In the OBSTACLE task, the robot elongates its body and leans forward in order to leverage the force of gravity to swiftly stride over the obstacle. Subsequently, the robot grows two leg-like stumps and uses them to walk, enabling efficient forward progress.

In the CATCH task, the robot initially alters its configuration to a slimmer form to access a confined space, followed by a transformation into an "L" shape to manipulate a square object to the target location. This demonstrates remarkable versatility when confronted with complex environmental conditions.

Lastly, for the SLOT task, the robot strategically reduces its size to maneuver through an exceptionally narrow space, resembling a mouse fitting into a small hole. To push the target cap atop a box, it then showcases its reconfigurability by regrowing its stature, providing the necessary height to manipulate the cap effectively.

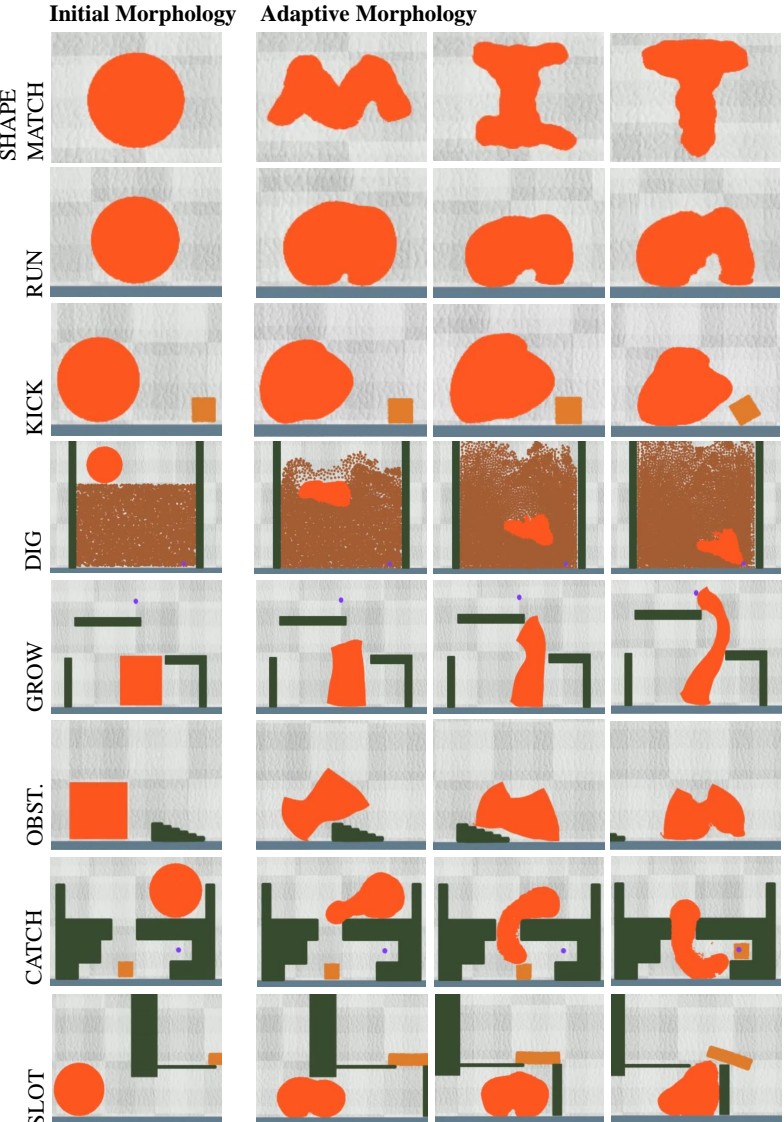

Figure 7: **DittoGym.** The visualization of robots successfully completing all eight tasks defined within the DittoGym while being controlled by the policy generated from CFP.

## D   METHOD ILLUSTRATION

The policy model's structure is a fully-convolutional framework. The input of the model is a multi-channel state image, which contains the information of the robot's shape and velocity (from x and y direction). The output of the model is a same-size, same-place action image, representing the strength in the action field. The illustration is shown in Figure 8 (in Figure 3, we use animation to illustrate the real input and output data). In our project, we employ a grid with a resolution of $64 \times 64 \times 2$ to store the action information. At every time step, the process involves two key stages:

- **Upsampling Coarse Action.** The coarse action, represented at a resolution of $8 \times 8 \times 2$, is first upscaled to match the $64 \times 64 \times 2$ grid. This upsampling is a vital step in preparing the action for distribution.

- **Distributing Signals to Particles.** Once upscaled, these action signals are then distributed to the individual particles in MPM's grid operation stage. This distribution is a critical component of how the robot receives and reacts to action commands.

The same methodology is applied when dealing with fine policy. The coarse action is firstly upsampled to $16 \times 16 \times 2$, then calculated the fine action using equation in Section 4.2, and finally to output the signals we upsample the result to $64 \times 64 \times 2$.

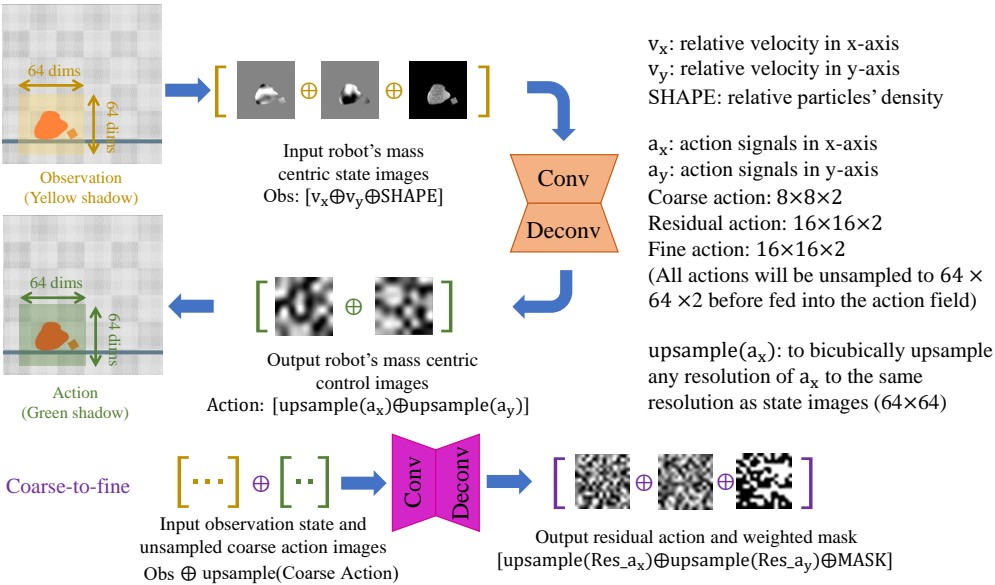

Figure 8: **Further illustration of CFP.**

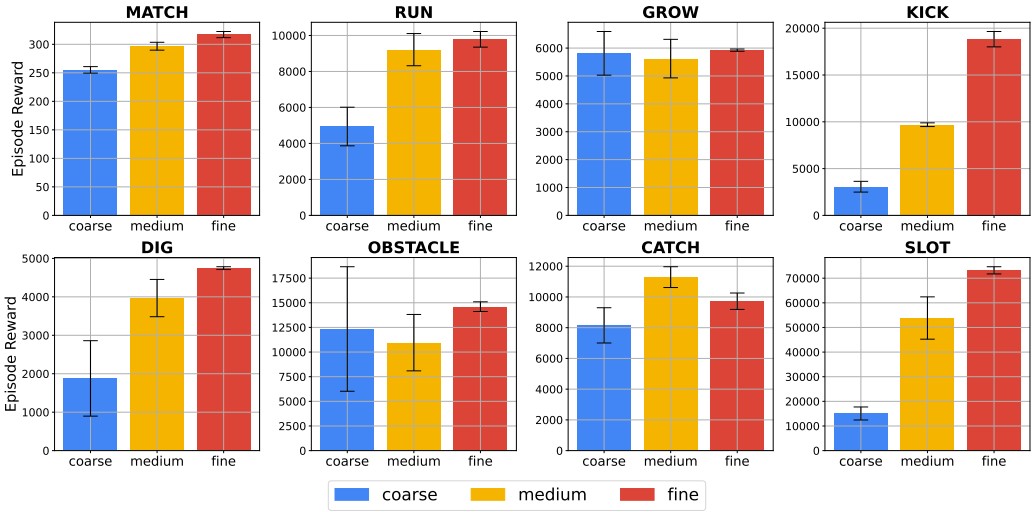

Figure 9: **Performance of expert policies under different action resolutions.** Expert policies using the highest resolution (fine) can achieve much higher episode rewards compared to those with medium or coarse resolutions, illustrating the tasks in DittoGym require *fine-grained* morphology changes.

# E    MODULAR-BASED BASELINES

The modular robot control methods are not directly applicable for our problem because highly-reconfigurable robots lack articulated joints, links and thus lack of explicit topological structure these baselines need. Despite that, we adopt the Modular Policy baseline (Pathak et al., 2019) by assigning topology with k-means clustering so GNN can be applied. We provide an additional baseline by letting an attention mechanism to figure out topology itself. Figure 10 illustrates the cluster while Figure 11 shows CFP's performance is still much higher than the dense-segmented modular baselines in most tasks (except for CATCH task).

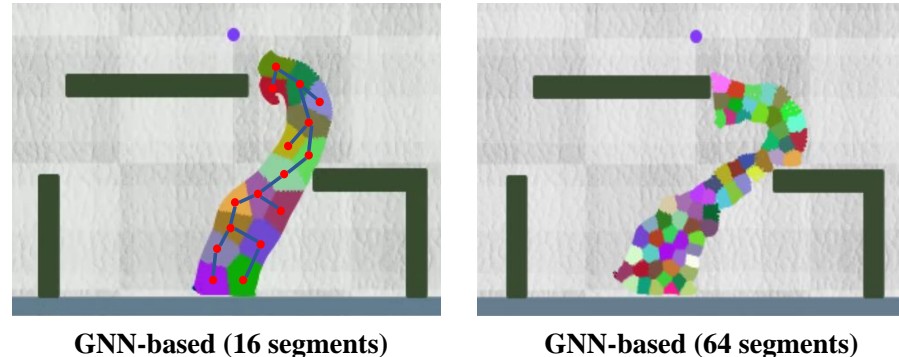

**GNN-based (16 segments)**          **GNN-based (64 segments)**

Figure 10: **Illustration of GNN-based modular control policy.** The robot's particles are segmented into several pieces (16 or 64) and then being treated as joint inputs to the modular policy. For GNN-based method, we need to artificially establish the graph among the segments according to their adjacent relationships, as shown on the left.

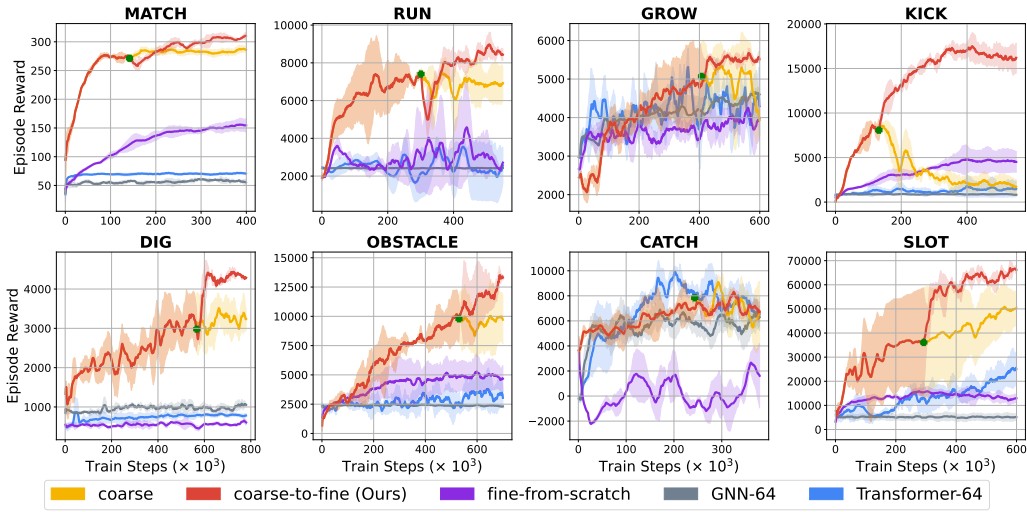

Figure 11: **Comparison of fine-resolution policies trained with and without coarse-to-fine technique.** In Figure 6, we compare the performance of CFP with sparse-segmented (16 segments) modular policies. In this section, we further compare its performance with dense-segmented (64 segments) modular policies.

# F    OTHER EVALUATION METRICS

We compare the running speed and the usage of memory of the proposed method with modular-based baselines in the RUN task. The results in Table 4 indicate that CFP is a faster and more lightweight method.

Table 4: **Running Speed and RAM Consumption Comparison.**

| Method | Average Time Per Episode (s) | Used RAM (MB) |
|---|---|---|
| **CFP** | 54 | 4053 |
| Transformer-based (dense-segment) | 104 | 6895 |
| GNN-based (dense-segment) | 83 | 4199 |

We report the success rate of the SLOT task whose success can be clearly defined. The results in Table 5 indicate that CFP outperforms the additional baselines.

Table 5: **SLOT Task Success Rate Comparison.**

| Method | SLOT (%) |
|---|---|
| **CFP** | 15.4 |
| Transformer-based (dense-segment) | 5.9 |
| GNN-based (dense-segment) | 0 |

We also conduct experiments on CFP to test if it is robust to the observation or actuation failure cases. we already added Gaussian noise to action output when getting our result in the main paper, to emulate noisy control. The SAC algorithm naturally optimizes expected return under high entropy (noisy) actions, which explains why our result is already good in the paper.

Our two added experiments randomly zeros out 20% of the action output or observation input respectively when evaluating our expert policy. This emulates actuator or sensor failures respectively. Figure 12 shows their performance as a ratio to original performance. Our performance is negligibly impacted (<10%) by action failure except 1 out of the 8 tasks, DIG, which shows a performance drop of only 30%. The policy also seems to show some robustness to sensor failure from the plot, a less common setting.

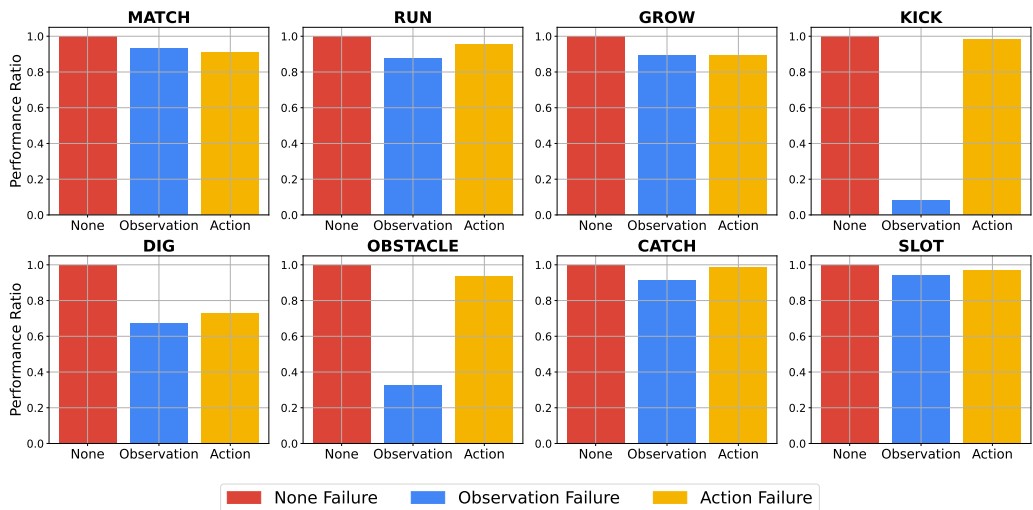

Figure 12: **Robustness justification for CFP.**

