# OpenReview forum: "DittoGym: Learning to Control Soft Shape-Shifting Robots"
_ICLR.cc/2024/Conference — ICLR 2024 poster_

### Official Review · Reviewer_UEFW · 2023-11-01

**Soundness:** 3 good
**Presentation:** 3 good
**Contribution:** 2 fair
**Rating:** 5
**Confidence:** 2

**Summary:**

This paper studies the problem of reconfigurable robots, which can change morphology to accomplish a task. The paper makes the following contributions: (a) a new simulator to model both elastic and plastic deformation; (b) a benchmark based on the aforementioned simulator, and (c) a coarse to fine reinforcement learning approach for controlling such robots.

**Strengths:**

1. The proposed simulator and benchmark fill a hole in the current literature on morphology-changing robots
2. The proposed hierarchical control approach outperforms other ablations and another baseline in a set of experiments.

**Weaknesses:**

Overall, I found this paper quite interesting to read. The main limitations, I think, are:
1. It is unclear whether the proposed simulator and benchmark is a good model of reality. Is it modeling a specific robot or class of robots? Could there be some experiments to show the validity of the simulator? I'm not necessarily referring to simulation to reality transfer, but rather to show that the simulator dynamics are correlated with the dynamics of a real robot.
2. The proposed hierarchical reinforcement learning approach is relatively standard. Its application to the problem of morphology-changing robots might be new, though.
3. I don't understand the reasoning behind the proposed baselines. There seem to be other approaches that are more related to the problem (e.g. Phatak et al., Learning to Control Self-Assembling Morphologies: A Study of Generalization via Modularity; Whitman et al., Learning Modular Robot Control Policies). Having a more detailed comparison could give a better idea of how difficult is the proposed benchmark for current algorithms.

Minor points:
1. Why are these tasks selected? What exactly is it that they evaluate? Why is it important to have them instead of others?
2. I am not sure I understand how the baseline of Neural Field Policy is implemented and whether this is a novel contribution of the paper or some other work proposed before for this problem.
3. I don't think Fig. 6 is helpful, the concept it represents is trivial and easily explained in a sentence

**Questions:**

It would be great if the authors could clarify why the proposed simulator is a good model of real robots, why the proposed tasks were selected, and clarify the experimental setup. In addition, it would be great to include more relevant benchmarks for this problem.

---

> ### Author Response · Authors · 2023-11-20
> **Rebuttal Part (1/2)**
>
> Thank you for your helpful comments. We respond to your comments below as well as adding more baselines.
>
> > It is unclear whether the proposed simulator and benchmark is a good model of reality. Is it modeling a specific robot or class of robots? Could there be some experiments to show the validity of the simulator? I'm not necessarily referring to simulation to reality transfer, but rather to show that the simulator dynamics are correlated with the dynamics of a real robot.
>
> Thank you for pointing out the real world connection! We took great caution throughout the project to make sure everything is governed by real-world physical equations. Our physical simulation is based on MPM, a widely used simulation algorithm that’s designed to simulate real world materials like liquids and plastics. Our robot material is simulated in a way governed by equations of the real stress-strain curve of plastic-elasto materials. The actuation is governed by that of magnetic field (which is used to control the first real-world highly reconfigurable robot [26]), and follows the original implementation in Taichi that simulates point gravity attraction. The only change we made is to normalize the actuation to follow conservation of momentum because we are parameterizing internal forces only. This is done before feeding actuation into simulation and doesn’t break any physical equation itself.
>
> > The proposed hierarchical reinforcement learning approach is relatively standard. Its application to the problem of morphology-changing robots might be new, though.
>
> Thanks for mentioning HRL, a nice potential addition to our related work. While our method has a spatial resolution hierarchy, it’s quite different from the hierarchy in HRL. In HRL, high-level policy and low-level policies are designed to be disentangled, such that they never share the input-output behavior. Rather, the high-level policy’s output is usually fed to low-level policy as input. For example, The famous D4RL [27] environment “ant maze” is widely used in HRL [28][29], where a high-level policy outputs local goals and low-level policy outputs actuations.
>
> In contrast, our policy of different resolutions shares the same 2D input and output space. The action field of high-resolution policy and that of low-resolution are directly added algebraically (The weighting mask and feed of coarse action into fine policy are optional). This is because of our hierarchy in the sense of spatial hierarchy, like CNNs and Feature Pyramids in computer vision, rather than the hierarchy in HRL. We appreciate the relation you pointed out and would add explanations to our paper upon acceptance.
>
> > I don't understand the reasoning behind the proposed baselines. There seem to be other approaches that are more related to the problem (e.g. Pathak et al., Learning to Control Self-Assembling Morphologies: A Study of Generalization via Modularity; Whitman et al., Learning Modular Robot Control Policies). Having a more detailed comparison could give a better idea of how difficult the proposed benchmark for current algorithms is.
>
> Thanks for bringing up these baselines. We’d like to clarify that suggested baselines aren’t directly applicable here while also providing results of the adopted version of suggested baseline [30].
>
> * Clarification: highly-reconfigurable robots lack articulated joints, links and thus lack of explicit topological structure these baselines need, as we highlighted in general response.
>
> * **Additional experiments**: Despite that, we adopt the suggested Modular Policy baseline[28] by assigning topology with k-means clustering so GNN can be applied. We provide an additional baseline by letting an attention mechanism to figure out topology itself. We benchmark the baselines on 4 of the 8 tasks due to time constraints with the same number of seeds.
>
> Details of both baselines and result curves are added to our [rebuttal website](https://sites.google.com/view/morphologicalmazerebuttal). Figure 1.1 in the website illustrates the cluster while figure 1.2 shows our method’s performance is still much higher than the added baselines. This is not surprising, as GNN is only suited for robots with clear articulation, which slime robots lack.

---

> > ### Comment · Reviewer_UEFW · 2023-11-23
> > **Thank you for your rebuttal**
> >
> > ```
> > While our method has a spatial resolution hierarchy, it’s quite different from the hierarchy in HRL. In HRL, high-level policy and low-level policies are designed to be disentangled, such that they never share the input-output behavior
> > ```
> > Thanks for clarifying this. I am, however, unsure why the term hierarchical was used. This is much more reminiscent of the RL technique of iterative skill learning, which is used for simulation to reality transfer (see, for example, Learning Locomotion Skills for Cassie: Iterative Design and Sim-to-Real by Xie et al, CoRL 2019).
> >
> > ```
> > Details of both baselines and result curves are added to our rebuttal website. Figure 1.1 in the website illustrates the cluster while Figure 1.2 shows our method’s performance is still much higher than the added baselines. This is not surprising, as GNN is only suited for robots with clear articulation, which slime robots lack.
> > ```
> > Thanks for adding the new results!
> >
> > Overall, I am happy to increase my score. However, I'll not go beyond the acceptance threshold since I am not convinced the real-world applicability or usefulness was well justified.

---

> ### Author Response · Authors · 2023-11-20
> **Rebuttal Part (2/2)**
>
> > Why are these tasks selected? What exactly is it that they evaluate? Why is it important to have them instead of others? Why the proposed tasks were selected, and clarify the experimental setup. In addition, it would be great to include more relevant benchmarks for this problem.
>
> Our task selection follows both generic RL benchmark design principles and the specific needs of reconfigurable robots. We highlight the factors we hoped to cover in our benchmark and the corresponding factor vector of each environment:
> * Tasks have to cover a range of difficulties (A=easy, B=hard)
> * Some tasks evaluates robot itself, others requires the robot’s ability to interact with external objects (A=not require, B=require)
> * Some tasks require just one morphology change, while others need changes of morphology multiple times (A=one change, B=multiple changes)
> * Some have non-convex reward landscape that requires algorithm to be long-horizon (A=short-horizon, B=long-horizon)
> * Some features softer material and some features less soft material (A=softer, B=less softer)
>
>   | Task        | Factor Vector     |
>   |-------------|-------------------|
>   | SHAPE_MATCH | [A, A, A, A, A]   |
>   | RUN         | [A, A, A, A, B]   |
>   | GROW        | [A, A, A, A, A]   |
>   | KICK        | [B, B, A, A, B]   |
>   | DIG         | [B, B, A, A, B]   |
>   | OBSTACLE    | [B, A, B, B, B]   |
>   | CATCH       | [B, B, B, B, A]   |
>   | SLOT        | [B, B, B, B, B]   |
>
>
>
> Each task is designed to represent certain factor combinations that we deem important, and we believe they together can give users a comprehensive understanding that a lot of RL ablations do through choices of tasks.
>
> We agree that more benchmarks would be helpful! However, our problem is the first of its kind. Reconfigurable robot benchmarks are not possible before us due to the lack of formalization and difficulty to make physical simulations, which is our main contribution. While we only have 8 environments in our paper, these two contributions make it much easier for the RL community to contribute more reconfigurable robot tasks. We plan to gradually add more tasks to the benchmark to achieve higher impact in the long term but we also believe that quality of benchmark is more significant in a first benchmark.
>
> > I am not sure I understand how the baseline of Neural Field Policy is implemented and whether this is a novel contribution of the paper or some other work proposed before for this problem.
>
> Sorry for the confusion. Neural Field Policy is designed to be a method that can take advantage of our infinitely dimensional action space by querying values in a coordinate-conditioned way. We recognize the confusion it caused, and will replace it in the camera-ready version with the two added baselines following your suggestion.
>
> > I don't think Fig. 6 is helpful, the concept it represents is trivial and easily explained in a sentence
>
> Thank you for your suggestion! While the concept is simple, Fig. 6 is also designed to be a **visualization** of the resulting shape of different policies, just like how RL paper also demonstrates suboptimal runs of baselines in their videos to give viewers more insights. We will add more visualizations of different tasks in the camera-ready version.
>
> ***
>
> [26] Sun et al. (2022). Reconfigurable magnetic slime robot: deformation, adaptability, and multifunction. Advanced Functional Materials, 32(26), 2112508.
>
> [27] Fu et al. (2020). D4rl: Datasets for deep data-driven reinforcement learning. arXiv preprint arXiv:2004.07219.
>
> [28] Li et al. (2022). Hierarchical planning through goal-conditioned offline reinforcement learning. IEEE Robotics and Automation Letters, 7(4), 10216-10223.
>
> [29] Nachum et al. (2019). Why does hierarchy (sometimes) work so well in reinforcement learning?. arXiv preprint arXiv:1909.10618.
>
> [30] Pathak et al. (2019). Learning to control self-assembling morphologies: a study of generalization via modularity. Advances in Neural Information Processing Systems, 32.

---

> ### Author Response · Authors · 2023-11-22
>
> Dear Reviewer UEFW,
>
> We first thank you again for your comments and suggestions.
>
> In our earlier response, we have provided detailed clarification based on your questions about the reality meanings of our proposed method and the settings of our benchmark. In addition, we illustrated the unique technique contributions of our method. Furthermore, we have conducted additional experiments with strong baselines to illustrate the effectiveness of our proposed method on our [rebuttal website](https://sites.google.com/view/morphologicalmazerebuttal) following your great suggestion. Finally, we have provided detailed analysis about the characteristics of tasks in our benchmark to illustrate the rationality of the task designs.
>
> As we are ending the stage of the author-reviewer discussion soon, we kindly ask you to review our revised paper and our response and reconsider the scores if our response has addressed your concerns.
>
> If you have any other questions, we are also pleased to respond. We sincerely look forward to your response.
>
> Best wishes!
>
> The authors

---

> ### Author Response · Authors · 2023-11-23
>
> Thank you for raising the score!
>
> >  why the term hierarchical was used
>
> Sorry for the confusion. While hierarchical is widely used in HRL, it's also a term in computer vision used widely in super-resolution and feature pyramid (anything that has multi-resolution stage or representations). We will try to avoid such conflict of terms in camera-ready.

---

### Official Review · Reviewer_mQFN · 2023-11-01

**Soundness:** 3 good
**Presentation:** 3 good
**Contribution:** 3 good
**Rating:** 8
**Confidence:** 4

**Summary:**

This paper presents a reinforcement learning approach able to control a reconfigurable and simulated soft robot that has to change its shape to perform different tasks. In addition to the novel RL algorithm, which first modifies the coarse structure of the robot and then more fine-grained details, the authors also introduce a new benchmark environment for reconfigurable soft robots. The presented control approach is compared to various baselines, which do not employ a coarse-to-fine control but instead use fine-grained control directly.

**Strengths:**

- Exciting research with a path to being employed on real robots such as ones based on ferromagnetic slime
- Interesting multi-scale muscle field control, where course and fine grain actions can affect the robots morphology
- Introduced a new interesting benchmark “Morphological Maze” that tests algorithms for their ability to perform morphological change

**Weaknesses:**

- A potential weakness of the approach, especially when it should ultimately be transferred to real robots, is that the control of the robot's shape do not happen based on local interactions, i.e. the employed controller has to take into account the complete shape of the robot and its environment
- Missing relevant literature on evolving soft robots and simulators (e.g.VoxCad). "Cheney, Nick, et al. "Unshackling evolution: evolving soft robots with multiple materials and a powerful generative encoding." ACM SIGEVOlution 7.1 (2014): 11-23." In fact, in this work, the authors use a CPPN-encoding which is the foundation for neural field models and should be mentioned.
- "However, these search-based zeroth-order optimization methods are computationally demanding and inefficient” - is this shown in the paper? It's a common argument but it would be good to have some references here backing it up.
- Importantly, what are the lessons for the larger machine learning community? e.g. for which other tasks could the course-to-fine (CFP) algorithm be useful? What about other hierarchical RL methods? The research is exciting from an alife/robot perspective but it could be better motivated for an ICLR audience
- Since this conference is about representation learning, it would be interesting to investigate further what type of representations the RL algorithms learn to control these soft robots. How do the coarse and fine-grained control interact to solve the tasks at hand?

**Questions:**

- Is the code available?
- How computationally expensive is the simulation environment?
- How large is the action space?
- how is the upsampling of coarse actions done?
- What are the network details (e.g. number of layers etc.?)

---

> ### Author Response · Authors · 2023-11-20
> **Rebuttal Part (1/2)**
>
> Thank you for your helpful comments. We respond to your comments below.
>
> > A potential weakness of the approach, especially when it should ultimately be transferred to real robots, is that the control of the robot's shape does not happen based on local interactions, i.e. the employed controller has to take into account the complete shape of the robot and its environment.
>
> Thank you for making the connection to real robots. First-person-view observation and third-person view observation are both common choices in real robot reinforcement learning papers. We focus on third-person-view exactly because the current real-world hardware implementations [20] [21] of the robot uses a third-person view setup. While we recognize that reconfigurable robot research may evolve in the long-run, we believe our setup and benchmark is best suited for current hardware.
>
> > Missing relevant literature on evolving soft robots and simulators (e.g.VoxCad). "Cheney, Nick, et al. "Unshackling evolution: evolving soft robots with multiple materials and a powerful generative encoding." ACM SIGEVOlution 7.1 (2014): 11-23."
>
> Thank you for the mention and we will add it to related work! However, we'd still like to emphasize that our problem is very different from co-design, as we explained in the general response.
>
> > However, these search-based zeroth-order optimization methods are computationally demanding and inefficient” - is this shown in the paper? It's a common argument but it would be good to have some references here backing it up.
>
> Thanks for your suggestion. We will add more references to support this claim in camera-ready.
>
> > Importantly, what are the lessons for the larger machine learning community? e.g. for which other tasks could the coarse-to-fine (CFP) algorithm be useful?
>
> Coarse-to-fine has seen its success in many machine learning domains like  image and video generation [22][23]. However, it’s been largely ignored by the RL community because not many RL problems largely ignore the structures of action space. The lesson is that whenever resolutional hierarchy exists, don’t overlook it just because this is not traditionally used in your domain.
>
> > Since this conference is about representation learning, it would be interesting to investigate further what type of representations the RL algorithms learn to control these soft robots. How do the coarse and fine-grained control interact to solve the tasks at hand?
>
> Thank you for defending the relevance of the conference. From the “About Us” page of ICLR, “The International Conference on Learning Representations (ICLR) is the premier gathering of professionals dedicated to the advancement of the branch of artificial intelligence called representation learning, but **generally referred to as deep learning**.”
>
> Therefore, as a deep reinforcement learning paper we naturally fit into the focus of ICLR.
>
> > Is the code available?
>
> Yes. We plan to release the code for both benchmark and algorithm upon acceptance.
>
> > How computationally expensive is the simulation environment?
>
> We made it really fast! The simulation is GPU based. We report the usage and speed of our most computation-intensive task DIG (because it contains the most particle numbers). Simulation alone, the environment runs at an average speed of 70.1 FPS on a RTX4090 GPU, with only 20-30% volatile utility usage and 15% memory usage. Therefore theoretically you can run 5 of such simulations on RTX4090 at similar speed.
>
> > How large is the action space?
>
> Although our formalization allows an infinite dimensional action space, our finest policy runs with an output of size 2*16*16 = 512 dims, which gets interpolated to this infinite dimensional action space.
>
> > How is the upsampling of coarse actions done? What are the network details (e.g. number of layers etc.?)
>
> Sorry about the confusion. The upsampling of coarse action is done by the bicubic interpolation while the network details are shown below:
>
> * Critic Network:
> The 3-layer CNN based encoder can embed the input of 5x64x64 (3 channels for state, 2 channels for the upsampled action) images into a vector with 512 dims following a CNN architecture widely used in RL papers and frameworks [24][25]. Then it will get through a 3-layer MLP with 256 dims latent space to yield the Q value.
> * Policy Network:
> The similar 3-layer CNN based encoder can embed the input of 3x64x64 images into a embedding shaped 32x4x4 (still 512 dims). Then it will get through a 3-layer Deconv to yield grid-like action (2x8x8 or 2x16x16).
>
> In the residual training stage, the policy network’s input channel will also be 5 (3 channels for state, 2 channels for the upsampled coarse action).

---

> > ### Comment · Reviewer_mQFN · 2023-11-20
> >
> > Thank you for the clarifications. Regarding my comment "it would be interesting to investigate further what type of representations the RL algorithms learn to control these soft robots. How do the coarse and fine-grained control interact to solve the tasks at hand?". I didn't doubt that the paper fits ICLR but nevertheless, it would still be interesting to investigate how the RL algorithms does what it does and what representations it did learn (at least for future work).
> >
> > It would be good if the model details were added to the paper to facilitate reproducing the results.

---

> > > ### Author Response · Authors · 2023-11-21
> > >
> > > > RL algorithms does what it does and what representations it did learn
> > >
> > > Thank you for your clarification! This does sound like a good ablation, as our action space has a 2D structure unlike previous works. We will try to analyze such 2D pattern like masks before the rebuttal deadline, but will definitely include visualizations in our camera-ready if we found them insightful.
> > >
> > > >  It would be good if the model details were added to the paper to facilitate reproducing the results.
> > >
> > > We will definitely do so! Hopefully our textual description above gives you more insight, and we also added an additional architecture figure on our [rebuttal website](https://sites.google.com/view/morphologicalmazerebuttal). We will also definitely release the entire code base for higher impact.
> > >
> > > We thank you again for your time, and would love to answer more of your questions! In the meanwhile, we added additional experiments in general response and wish they can provide you with more insights.

---

> > > ### Comment · Reviewer_mQFN · 2023-11-22
> > >
> > > Given that my questions were answered, I'm happy to increase my score.

---

> > > > ### Author Response · Authors · 2023-11-22
> > > >
> > > > Thank you! We will make sure updates are organized and updated to the paper in our rebuttal revision or camera-raedy.

---

> ### Author Response · Authors · 2023-11-20
> **Rebuttal Part (2/2)**
>
> [20] Jangir et al. (2022). Look closer: Bridging egocentric and third-person views with transformers for robotic manipulation. IEEE Robotics and Automation Letters, 7(2), 3046-3053.
>
> [21] Zhang et al. (2019, May). Solar: Deep structured representations for model-based reinforcement learning. In International conference on machine learning (pp. 7444-7453). PMLR.
>
> [22] Cho et al. (2021). Rethinking coarse-to-fine approach in single image deblurring. In Proceedings of the IEEE/CVF international conference on computer vision (pp. 4641-4650).
>
> [23] Dai et al. (2023). Emu: Enhancing image generation models using photogenic needles in a haystack. arXiv preprint arXiv:2309.15807.
>
> [24] Mnih et al. (2013). Playing atari with deep reinforcement learning. arXiv preprint arXiv:1312.5602.
>
> [25] Raffin et al. (2019). Stable baselines3.

---

### Official Review · Reviewer_NgHH · 2023-11-02

**Soundness:** 2 fair
**Presentation:** 3 good
**Contribution:** 2 fair
**Rating:** 5
**Confidence:** 4

**Summary:**

This paper proposes an approach to learning morphology for soft robot control. The paper formulates reconfigurable soft robot control as a high-dimensional reinforcement learning problem in a continuous 2D muscle field and designs a coarse-to-fine hierarchical policy (CFP) to expedite the exploration of the action space. The paper also implements a benchmark that allows simulating the plastic deformation of robots for various tasks.

**Strengths:**

+ Existing methods are well-reviewed, and the classification of existing challenges in reconfigurable soft robots is a strength.

+ The proposed benchmark is non-trivial, and the demo video supports the proposed method.

+ If applicable to real robots, the problem of controlling reconfigurable soft robots is important.

**Weaknesses:**

- The explanation of the full approach is unclear. For example, multiple variables and terms are not defined or explained in the figures, including vx, vy, and SHAPE in Figure 3, the dimension of actions, architectures of the encoder, Coarse, and Residual policy. In the appendix, the paper briefly explains that the core framework is based on existing works SAC and Nature CNN with minor modifications, but it is unclear how to reproduce the full approach.

- Given the concern above, the theoretical novelty seems not high, as the main theoretical novelty of CFP is the introduction of adding residual action to coarse action.

- The paper lacks comparisons with existing state-of-the-art methods. The experiments only evaluate the performance of two ablation baseline methods and one NFP method. More comparisons will be appreciated. Besides the evaluation metric on reward, if more metrics can be used to evaluate real-robot applications, it will be appreciated, such as the successful rate or time efficiency.

- As one of the main challenges proposed to address in this paper, the justification of lifetime adaptation is insufficient in the experiments. For example, if the approach can adapt to noise or actuator failures during the lifetime operation?

- Even though the simulation experiments are impressive, there is still a gap between applying the proposed work to real-world robots, as shown in the demo video. An explanation of how to extend the work from simulations to real-world robots will strengthen the paper significantly.

**Questions:**

Please see the weaknesses section.

---

> ### Author Response · Authors · 2023-11-20
> **Rebuttal Part (1/2)**
>
> Thank you for your helpful comments. We respond to your comments below.
>
> > The explanation of the full approach is unclear. For example, multiple variables and terms are not defined or explained in the figures, including vx, vy, and SHAPE in Figure 3.
>
> Sorry for the confusion. Our vx and vy are tracked velocities of particles, rasterized to pixels. SHAPE is the rasterized occupancy of particles. We will update the paper accordingly in the camera-ready version.
>
> > What is the dimension for actions?
>
> As we mentioned in section 5.2, the action output is 16x16x2 before interpolation, although the action space itself is infinite dimensional in our formalization.
>
> > What are the architectures of encoder, Coarse, Residual policy?
>
> Sorry about the confusion. We follow the natural CNN architecture widely used in RL. The network details are shown below:
> * Critic Network:
> The 3-layer CNN based encoder can embed the input of 5x64x64 (3 channels for state, 2 channels for the upsampled action) images into a vector with 512 dims following a CNN architecture widely used in RL papers and frameworks [11][12]. Then it will get through a 3-layer MLP with 256 dims latent space to yield the Q value.
> * Policy Network:
> The similar 3-layer CNN based encoder can embed the input of 3x64x64 images into a embedding shaped 32x4x4 (still 512 dims). Then it will get through a 3-layer Deconv to yield grid-like action (2x8x8 or 2x16x16).
>
> In the residual training stage, the policy network’s input channel will also be 5 (3 channels for state, 2 channels for the upsampled coarse action).
>
> We have added more detailed illustrations of our method in figure 2 on the [rebuttal website](https://sites.google.com/view/morphologicalmazerebuttal).
>
> > In the appendix, the paper briefly explains that the core framework is based on existing works SAC and Nature CNN with minor modifications, but it is unclear how to reproduce the full approach.
>
> We will open-source the entire code base upon acceptance. We will also update the paper to include these details.
>
> > Given the concern above, the theoretical novelty is not high, as the main theoretical novelty of CFP is the introduction of adding residual action to coarse action.
>
> While our paper is not a ML theory paper, we still believe we have strong technical contributions that can inspire broader research from the community. First, formalize a novel control problem that has never been studied algorithmically into a Markov Decision Problem. This establishes a platform that future researchers can build on. To our knowledge, we are the first paper to let robots control its muscle to change its own morphology, not to mention the slime robot. Secondly, we overcome the challenges of simulation and provide the RL community with a set of fast benchmarks of this interesting but novel task. Our effort on the simulation is non-trivial and such novel benchmarks alone have been traditionally regarded as technical contributions in many top ML conferences [13] [14] [15]. Thirdly, CFP is one of the first RL algorithms that uses a 2D action space. Many design choices here such as fully convolutional network or coarse to fine are largely overlooked by the RL community, which traditionally focuses on unstructured action space while our insight brings attention to such 2D structure. We believe our formalization, benchmarks and algorithm together shall constitute technical contributions above the acceptance threshold.
>
> > The paper lacks comparisons with existing state-of-the-art methods.
>
> Thanks for your suggestion. We added two more related baselines and corresponding experiments following Reviewer UEFW’s suggestion, although we’d like to clarify that these baselines have to undergo significant adoption as we have a very novel task.
>
> We take state-of-the-art algorithms in the field of articulated reconfigurable robots, Modular Policy [16],  and adopt it for highly-reconfigurable robots by using k-means to assign topological structures to robot muscles. We provide an additional baseline by letting an attention mechanism to figure out topology itself. We benchmark the baselines on 4 of the 8 tasks due to time constraints with the same number of seeds.
>
> Details of both baselines and result curves are added to our [rebuttal website](https://sites.google.com/view/morphologicalmazerebuttal). Figure 1.1 in the website illustrates the cluster while figure 1.2 shows our method’s performance is still much higher than the added baselines. This is not surprising, as GNN is only suited for robots with clear articulation, which slime robots lack.

---

> ### Author Response · Authors · 2023-11-20
> **Rebuttal Part (2/2)**
>
> > Besides the evaluation metric on reward, if more metrics can be used to evaluate real-robot applications, it will be appreciated, such as the successful rate or time efficiency.
>
> Thank you for the feedback. First we’d like to clarify that our provided metric, maximum episode reward and sample efficiency curves are usually regarded as the most fundamental metrics in the DRL community. In fact, a lot of highly cited DRL papers choose sample efficiency curve as the only metric [17][18]. This is because metrics like success rate can only be defined in tasks like manipulation, while a task like running forward doesn’t clearly have a threshold called “success”. Instead, designers may only want the robot to run as far as possible instead of letting it stop at a point.
>
> Despite this, we report the success rate of some manipulation or reaching tasks whose success can be clearly defined. In addition, we also provide the wall-clock-time metric for reference. In figure 3 of our [rebuttal website](https://sites.google.com/view/morphologicalmazerebuttal)), we report these metrics of our algorithm & baselines in two challenging manipulation tasks. We hope such information can give you readers more insight and we are happy to provide additional results if you have additional questions!
>
> > As one of the main challenges proposed to address in this paper, the justification of lifetime adaptation is insufficient in the experiments. For example, if the approach can adapt to noise or actuator failures during the lifetime operation?
>
> Thanks for your suggestion. We follow your suggestion and add 2 additional sets of experiments that emulate actuator failure and observation failure.
>
> Regarding noise adaptation, we’d like to clarify that we already added gaussian noise to action output when getting our result in the main paper, to emulate noisy control. The SAC algorithm naturally optimizes expected return under high entropy (noisy) actions, which explains why our result is already good in the paper.
>
> Our two added experiments randomly zeros out 20% of the action output / observation input respectively when evaluating our expert policy. This emulates actuator / sensor failures respectively. Figure 4 on our [rebuttal website](https://sites.google.com/view/morphologicalmazerebuttal) shows their performance as a ratio to original performance. As shown in the figure, our performance is negligibly impacted (<10%) by action failure except 1 out of the 8 tasks, DIG, which shows a performance drop of only 30%. The policy also seems to show some robustness to sensor failure from the plot, a less common setting.
>
> > Even though the simulation experiments are impressive, there is still a gap between applying the proposed work to real-world robots, as shown in the demo video. An explanation of how to extend the work from simulations to real-world robots will strengthen the paper significantly.
>
> Thanks for your comment. In our project, we tried to mimic the settings of real-world slime robots implementation in [19]. In this paper, the researchers use cameras and magnetic fields to demonstrate basic movements of the slime robot. In our benchmark, we adopt the gravity field simulation from Taichi to simulate magnetic fields and adopt the same real-world robot observation hardware researchers use. While we recognize the current framework has its assumptions, we believe our implementation is well-aligned with the hardware progress of real-world magnetic robots, which are bottlenecked by algorithms and benchmarks to which our paper aims to contribute.
>
> ***
>
> [11] Mnih et al. (2013). Playing atari with deep reinforcement learning. arXiv preprint arXiv:1312.5602.
>
> [12] Raffin et al.(2019). Stable baselines3.
>
> [13] Huang et al. (2021). Plasticinelab: A soft-body manipulation benchmark with differentiable physics. arXiv preprint arXiv:2104.03311.
>
> [14] Bhatia et al. (2021). Evolution gym: A large-scale benchmark for evolving soft robots. Advances in Neural Information Processing Systems, 34, 2201-2214.
>
> [15] James et al. (2020). Rlbench: The robot learning benchmark & learning environment. IEEE Robotics and Automation Letters, 5(2), 3019-3026.
>
> [16] Pathak et al. (2019). Learning to control self-assembling morphologies: a study of generalization via modularity. Advances in Neural Information Processing Systems, 32.
>
> [17] Hafner et al. (2019). Dream to control: Learning behaviors by latent imagination. arXiv preprint arXiv:1912.01603.
>
> [18] Laskin et al. (2020). Reinforcement learning with augmented data. Advances in neural information processing systems, 33, 19884-19895.
>
> [19] Sun et al. (2022). Reconfigurable magnetic slime robot: deformation, adaptability, and multifunction. Advanced Functional Materials, 32(26), 2112508.

---

> ### Author Response · Authors · 2023-11-22
>
> Dear Reviewer NgHH,
>
> We first thank you again for your comments and suggestions.
>
> In our earlier response, we have provided detailed clarification based on your questions about the method details. In addition to that, we also added additional method details in appendix section E and will release the code to make this fully reproducible.
>
> We have also conducted additional experiments with strong baselines to illustrate the effectiveness of our proposed method on our [rebuttal website](https://sites.google.com/view/morphologicalmazerebuttal) and we also compared the performance loss when faced with observation or action failures following your great suggestion. Furthermore, we have also listened to your useful advice to put up more metrics such as time efficiency and success rate to comprehensively evaluate our method. Finally, we have provided illustrations about how we can extend our project with real-world settings.
>
> As we are ending the stage of the author-reviewer discussion soon, we kindly ask you to review our revised paper and our response and reconsider the scores if our response has addressed your concerns.
>
> If you have any other questions, we are also pleased to respond. We sincerely look forward to your response.
>
> Best wishes!
>
> The authors

---

### Official Review · Reviewer_5Sv3 · 2023-11-02

**Soundness:** 3 good
**Presentation:** 2 fair
**Contribution:** 3 good
**Rating:** 6
**Confidence:** 2

**Summary:**

This paper proposes a new task of **reconfigurable** soft robot design. The goal of this task is to **continuously** optimize the morphology of soft robots **during** the episodes of accomplishing a long-horizon task.

This paper first develops a simulation platform that implements the action space and transition dynamics of soft robots and environments. It also implements diverse tasks for benchmarking. Specifically, they define long-horizon tasks that require multiple times of morphology changes.

To address the reconfigurable robot design task, this paper proposes a novel RL algorithm that enables morphology changes from coarse to fine. This algorithm leads to efficient morphology optimization.

This paper conducts experiments on the proposed simulator with 4 types of tasks.

**Strengths:**

1. The task of reconfigurable robot design is important. This paper formulates this task as an MDP and provides a simulator that implements the action space & transition dynamics of the MDP.
2. This paper proposes a novel residual RL algorithm that shows impressive results in the 4 types of tasks.

**Weaknesses:**

1. There is no adequate comparison with classical robot design baselines, e.g., Bayesian optimization, genetic algorithms, etc.

**Questions:**

1. Is the action space or environment 2D or 3D? Why use 2D images as raw states for policy instead of using ground truth robot states(i.e., exact morphology parameterizations)?

---

> ### Author Response · Authors · 2023-11-20
>
> Thank you for your helpful comments. We respond to your comments below.
>
> > There is no adequate comparison with classical robot design baselines, e.g. Bayesian optimization, genetic algorithms, etc.
>
> Thanks for your suggestion about the baselines. While co-design is closely related in our related work section, we’d like to clarify that we are working on a novel problem that is drastically **different** from co-design, rendering the co-design baselines unusable. As we mentioned in the general response, co-design problems **fix** a robot morphology during execution of the task once the algorithm finds an optimal morphology. In contrast, the goal of our project is to control a type of robot that does **not** have a fixed morphology. In our problem, the slime robot has to actively control and **change** its morphology throughout the task. In fact, our robot always begins with a circle or square shape in any task.
>
> Despite this, we added two more related baselines and corresponding experiments following Reviewer UEFW’s suggestion. We take state-of-the-art algorithms in the field of articulated reconfigurable robots, Modular Policy [4],  and adopt it for highly-reconfigurable robots by using k-means to assign topological structures to robot muscles. We provide an additional baseline by letting an attention mechanism to figure out topology itself. We benchmark the baselines on 4 of the 8 tasks due to time constraints with the same number of seeds.
>
> Details of both baselines and result curves are added to our [rebuttal website](https://sites.google.com/view/morphologicalmazerebuttal). Figure 1.1 in the website illustrates the cluster while figure 1.2 shows our method’s performance is still much higher than the added baselines. This is not surprising, as GNN is only suited for robots with clear articulation, which slime robots lack.
>
> > Is the action space or environment 2D or 3D?
>
> The action space is represented as a 2-D grid, although our simulator can be easily extended to 3D.
>
> > Why use 2D images as raw states for policy instead of using ground truth robot states(i.e., exact morphology parameterizations)?
>
> We choose image observations because pixels are the most general representation of tasks. e.g. Millions of video games are played by humans through pixels; typical real robot manipulation tasks in RL literatures. This has been widely recognized by the deep reinforcement community [5] [6], which considers control from pixels as a more challenging problem [7] [8].
>
> A second reason we consider pixel observation is its potential for real-world transfer. Slime robots, for example the real-world implementation [9] [10], are highly unstructured and hard to obtain an intrinsic state. Pixel observation not only addresses the challenge for us but also organically capturess external world.
>
> ***
>
> [4] Pathak et al. (2019). Learning to control self-assembling morphologies: a study of generalization via modularity. Advances in Neural Information Processing Systems, 32.
>
> [5] Mnih et al. (2013). Playing atari with deep reinforcement learning. arXiv preprint arXiv:1312.5602.
>
> [6] Shah et al. Rrl: Resnet as representation for reinforcement learning. arXiv preprint arXiv:2107.03380.
>
> [7] Laskin et al. (2020, November). Curl: Contrastive unsupervised representations for reinforcement learning. In International Conference on Machine Learning (pp. 5639-5650). PMLR.
>
> [8] Laskin et al. (2020). Reinforcement learning with augmented data. Advances in neural information processing systems, 33, 19884-19895.
>
> [9] Sun et al. (2022). Reconfigurable magnetic slime robot: deformation, adaptability, and multifunction. Advanced Functional Materials, 32(26), 2112508.
>
> [10] Wang et al. (2023). Reconfigurable Liquid‐Bodied Miniature Machines: Magnetic Control and Microrobotic Applications. Advanced Intelligent Systems, 2300108.

---

> > ### Comment · Reviewer_5Sv3 · 2023-11-21
> >
> > Thanks for the clarification!
> >
> > >The action space is represented as a 2-D grid, although our simulator can be easily extended to 3D.
> >
> > I'm not familiar with what work should be done to extend the 2D soft robot simulator to 3D. Could the author provide more details? Because I think 3D robots and environment simulations are significantly more realistic than 2D ones, it would be more beneficial to claim directly that this simulator can be extended to 3D and how to.
> >
> > Also, a detailed comparison with the current soft robot simulator would be appreciated. Similar to the one in Table 5 Page15: https://arxiv.org/pdf/2303.09555.pdf
> >
> >
> > >We choose image observations because pixels are the most general representation of tasks. e.g. Millions of video games are played by humans through pixels; typical real robot manipulation tasks in RL literatures. This has been widely recognized by the deep reinforcement community [5] [6], which considers control from pixels as a more challenging problem [7] [8].
> >
> > I'm kind of not convinced by this reason. Visual RL is important, but I didn't see the rationale of using pixel observation for robot design. I think robot parameterization is much more essential for robot design. If the exact state is hard to obtain in the real world (I admit that it is hard in real world), I'm wondering to what extent using pixel observation could affect the task formulation/difficulty, i.e., would you provide results of your proposed model&baselines that use accurate robot parameterization as the state? Since everything is in simulation, I suppose retrieve intrinsic robot parameterization from simulation is not difficult.
> >
> > >A second reason we consider pixel observation is its potential for real-world transfer. Slime robots, for example the real-world implementation [9] [10], are highly unstructured and hard to obtain an intrinsic state. Pixel observation not only addresses the challenge for us but also organically capturess external world.
> >
> > 2D pixel observation can be ambiguous in 3D environment. I'm wondering what type of observations should be adopted if extend this simulation to 3D?

---

> ### Author Response · Authors · 2023-11-21
>
> Thank you for your feedback! We sincerely wish our written justification provides additional insight to you in case we cannot get additional results done in time.
>
> > extend the 2D soft robot simulator to 3D ... more beneficial to claim directly that this simulator can be extended to 3D and how to
>
> Thank you for your question and suggestions! Our physics simulation is coded with [Taichi](https://github.com/taichi-dev/taichi), which simulates 2D and 3D alike with same equations and algorithms (MPM). A widely used piece of sample code for 3D MPM simulation can be found [here](https://github.com/hzaskywalker/PlasticineLab/blob/main/plb/engine/mpm_simulator.py), (it also demonstrates how to make taichi mpm differentiable) which shares the same material equations as [2D code sample](https://github.com/hzaskywalker/PlasticineLab/blob/main/plb/engine/mpm_simulator.py) except you change things like 2D rotation matrix to 3D and change 2 loops from 3 loops; even the variable names are shared. One can put these two files side-by-side and translate our simulation in 3D without much problem.
>
> > detailed comparison with the current soft robot simulator
> We provide the suggested table below. Since we are not a co-design benchmark like the one you mentioned and we cited, our environments uniquely allows and requires robots to actively change their shape to accomplish a task (initial shape is always sphere or cube). We think it's an amazing suggestion will include it in our camera ready.
>
> | Platform                       | Simulation Method    | Tasks                      |  Multiphysical Materials                | Morphology Changes During Task | Morphology Changes Need Active Control  | Fine-grained Morphology Changes |  Differentiability|
> |:--------------------------------:|:----------------------:|:----------------------------:|:-----------------------------------------:|:---------------------------------:|:-----------------------------------------:|:------------------:|:------------------:|
> | SoMoGym (Graule et al., 2022) | Rigid-link System     | Mostly Manipulation        |                                         |                                  |                                        |                   |                |
> | DiffAqua (Ma et al., 2021)    | FEM                   |  Swimmer                   |                                          |                                  |                                       |        ✓          |          ✓          |
> | EvoGym (Bhatia et al., 2021)  | 2D Mass-spring System | Locomotion & Manipulation  |                                          |                                  |                                       |                   |                  |
> | SoftZoo (Wang et al., 2023)   | MPM                   | Locomotion                 |                    ✓                      |                                 |                                       |         ✓           |      ✓           |
> | Morphological Zoo (Ours)      | MPM                   | Locomotion & Manipulation  |                     ✓                     |                  ✓               |                  ✓                   |         ✓           |                 |
>
> > Visual RL is important, but I didn't see the rationale of using pixel observation for robot design
>
> This is because we are not doing robot design, but learning to control a new type of robot! The reason why early RL researchers use state is that control from pixel is hard [7][8], and early algorithms can hardly work unless one gives it the state. From 2018, with the emergence of a good visual-motor-control algorithm, RL community and robot learning researchers have started criticizing state-observation as they can hardly transfer to real world [5] [6] [7] [8].
>
> On the other hand, this was not a problem for co-design community, many of whom also do computer graphics and material science, fields which assume materials can be simulated and analyzed in simulators (and often differentiable). Under such assumption, one use computer algorithms to optimize for a design in a simulator before one can 3D print or manufacture a robot. However, to our best-knowledge, such robots cannot be controlled in real world due to state-spaced policy.
>
> While we recognize the merit of both school of thoughts in the problems they care about, our goal aligns better with RL & robot learning community.
>
> We will try our best to run state-based baselines before rebuttal deadline, though it may take a while to train enough seeds. Still, many important works in RL [5][6][7][8] point out that control from image observation is much harder and useful for real-world.
>
> > 2D pixel observation can be ambiguous in 3D environment
>
> Good question. To solve this, multi-camera observation has become standard in real-robot robot learning research, especially for robotics manipulation. For example, a popular [imitation learning method](https://arxiv.org/abs/2304.13705).

---

> > ### Comment · Reviewer_5Sv3 · 2023-11-21
> >
> > >Our physics simulation is coded with Taichi, which simulates 2D and 3D alike with same equations and algorithms (MPM). A widely used piece of sample code for 3D MPM simulation can be found here, (it also demonstrates how to make taichi mpm differentiable) which shares the same material equations as 2D code sample except you change things like 2D rotation matrix to 3D and change 2 loops from 3 loops; even the variable names are shared. One can put these two files side-by-side and translate our simulation in 3D without much problem.
> >
> > Thanks for providing those contexts!
> >
> > >detailed comparison with the current soft robot simulator We provide the suggested table below.
> >
> > Would you like to explain the lack of **Differentiability**? And what does it mean for the proposed task formulation (which aspect and to what extent)?
> >
> > >We will try our best to run state-based baselines before rebuttal deadline, though it may take a while to train enough seeds.
> >
> > Thanks for considering adding those results! It will be helpful to understand how much extent using third-person pixel observations would affect the task properties, especially when the task itself is a novel contribution.
> >
> > Also, how are the third-person visual observations obtained? Are they from a fixed camera or a moving camera?
> >
> > >Still, many important works in RL [5][6][7][8] point out that control from image observation is much harder and useful for real-world.
> >
> > I acknowledge the importance of control on visual observations in some robotic tasks. On the other hand, I think for this task, it's beneficial to understand the influence of different types of observation on the task itself and the potential future methods.

---

> > > ### Author Response · Authors · 2023-11-22
> > >
> > > Dear reviewer,
> > >
> > > > Differentiability
> > >
> > > The lack of our differentiability is from our motivation.
> > >
> > > Differentiable physics is widely used by computer graphics and robot co-design, because the ability to simulate materials accurately is a core assumption of these fields. In RL, people don't make this assumption because real-world often has non-state observations, and robots has to interact with external world which breaks differentiability. Even worse, using differential simulation is actually bad for policy learning in contact-rich dynamics, as pointed out by the [ICML 2022 outstanding paper](https://arxiv.org/abs/2202.00817) because they don't do smooth contact dynamics.
> > >
> > > In fact, in the above table, only environments without manipulation tasks feature differentiable simulators while our benchmark features a lot of manipulation or contact rich tasks. We also have an objective that better aligns with RL community and roboticist, using more general observations and doing a lot of interactions with external world.
> > >
> > > On the other hand, we recognize that our code can be modified to do co-design as well so one may want differentiability. Although not provided, this is not excessively hard either, as I pointed out when I mention the 3D mpm code base, which serves as a good example of how to add differentiability. In our code, we implemented all physical equations, and we simply need to double each line and implement the gradient equation as well.
> > >
> > > > how are the third-person visual observations obtained?
> > >
> > > This a moving camera, although in real world one may use a single camera followed by cropping because real-world slime robots are very small
> > >
> > > > understand the influence of different types of observation
> > >
> > > Our lately added GNN Modular Policy baseline on our [rebuttal website](https://sites.google.com/view/morphologicalmazerebuttal) uses state space observation. It is the state-of-the-art method to leverage structure of state space in a closely related problem. As you can see from figure 1.2, this baseline method is actually much lower than us.
> > >
> > > Please don't hesitate to ask us additional questions! I'd also really appreciate it if you could consider lifting your score based on the rebuttal.

---

> ### Comment · Reviewer_5Sv3 · 2023-11-22
>
> >The lack of our differentiability is from our motivation.
>
> Thanks for providing the helpful context and references!
>
> >Our lately added GNN Modular Policy baseline on our rebuttal website uses state space observation. It is the state-of-the-art method to leverage structure of state space in a closely related problem. As you can see from figure 1.2, this baseline method is actually much lower than us.
>
> I think the more important comparisons are:
> 1. your model with accurate state vs. your model with pixel observation;
> 2. baselines with accurate state vs. baselines with pixel observation.
>
> Because those comparisons are more for understanding the problem setting than for your method justification.

---

> ### Author Response · Authors · 2023-11-22
>
> Dear reviewer,
>
> We will try our best to include these results.
>
> We really appreciate the time you spent on understanding our work. If you don't feel comfortable enough raising the scores, please consider raise your confidence for our clarifications!

---

> > ### Comment · Reviewer_5Sv3 · 2023-11-22
> >
> > All my concerns are addressed.
> > The remaining two comparisons mentioned above will be helpful and appreciated.
> >
> > I maintain my scores as Rating: 6 and Confidence: 2.

---

### Author Response · Authors · 2023-11-20
**General Response**

We thank the reviewers for their comments and suggestions. We are pleased that the reviewers find our paper novel (Reviewers 5Sv3, mQFN), possess potential for realistic applications (Reviewers NgHH, mQFN) and find our benchmark  comprehensive and interesting (Reviewers 5Sv3, NgHH, mQFN, UEFW).

The outstanding concerns from reviewers centers around lack of baselines and confusion about our problem setting. Here we clarify our problem and highlight 2 out of our 4 experiment additions. We then address individual reviewer concerns in each reviewer response. Our goal is to control a type of robot that does **not** have a fixed morphology **nor** articulated links during task execution, just like the liquidmetal robot in the movie Terminator. Such highly-reconfigurable robots have been invented in the real world [1] that sees applications in medication but currently lacks appropriate simulations and algorithms. Such desired capabilities make the problem **very different** from the problem of co-design and modular robot control for the following reasons:

* Different from co-design problem (Reviewers 5Sv3), highly-reconfigurable robots actively **changes** its morphology **during** a task under an expert policy, while co-designed robots use a **fixed** morphology to accomplish tasks once the morphology is optimized. In addition, we need to learn the muscle actuation that’s needed to change robot morphology to accomplish tasks, while most co-design literatures directly initialize a robot to be the designed shape.

* Different from modular robots (Reviewer UEFW), highly-reconfigurable robots are NOT defined by a collection of rigid-body links and joints. It thus lacks a clear topological structure that GNN-based modular control methods typically rely on.

Due to such fundamental differences in the problem, many of the suggested co-design baselines such as Bayesian Optimization and Genetic Algorithm [2] cannot be used on reconfigurable robots, as we are **controling** morphological changes by muscles rather than **searching for** a mophology.

Although modular robot control algorithms are also not best-aligned with our problem out-of-the-box, we found a way to adapt the state-of-the-art baselines for our problem. We add 2 new strong baselines and present detailed results in our response. We first take Modular Policy [3] suggested by (Reviewer UEFW), a popular modular robot control algorithm. We artificially cluster the robot’s component particles into several muscle clusters with K-means, creating a topological structure which GNN policies can be applied on and run the suggested algorithm on it. A second baseline is added to avoid the limitation of adjacency assignment, by adding an attention-based architecture to Modular Policy [3], allowing it to figure out modular structures itself. We provide additional clarifications, explanations and discussion in the per-reviewer responses as well as our [**rebuttal website** here](https://sites.google.com/view/morphologicalmazerebuttal).

***

[1] Sun et al. (2022). Reconfigurable magnetic slime robot: deformation, adaptability, and multifunction. Advanced Functional Materials, 32(26), 2112508.

[2] Bhatia et al. (2021). Evolution gym: A large-scale benchmark for evolving soft robots. Advances in Neural Information Processing Systems, 34, 2201-2214.

[3] Pathak et al. (2019). Learning to control self-assembling morphologies: a study of generalization via modularity. Advances in Neural Information Processing Systems, 32.

---

> ### Comment · Reviewer_5Sv3 · 2023-11-22
>
> > Due to such fundamental differences in the problem, many of the suggested co-design baselines such as Bayesian Optimization and Genetic Algorithm [2] cannot be used on reconfigurable robots, as we are **controlling** morphological changes by muscles rather than **searching for** a mophology.
>
> I'm confused about the essential differences between controlling morphological changes and searching for a mophology.
>
> My understanding of **controlling** morphological changes is continuously searching for optimal morphologies. Can the former one be considered as a continuous loop of the latter one?

---

> ### Author Response · Authors · 2023-11-22
>
> Glad to make an important clarification here.
>
> Searching for a mophology refers to the design process of co-design. Once your algorithm finds out a design for a robot, at the frame 1 of your task, your robot is already in that shape. On the other hand, in our problem setting, our robot always have a shape of sphere in frame 1, and the robot need to figure out a sequence of muscle actions to bend it into a target shape.
>
> Imagine you are making sculptures of animals. Search for a mophology is like, you imagine the a shape in your mind and that's it.
> Controlling the mophology is like, you not only need to imagine the shape, but also need to use your hand to lift tools, infuse water, mix up the clay and take all these physical actions to create that sculpture.
>
> Some co-design papers also talks about control, which is something after you make the sculpture (e.g. you are making sculptures of animals and you want to evaluate whether the animal can run).  But their design itself has the difference we mentioned above. The control in co-design paper assumes someone already magically made the sculpture for you and then you do something on top of it, which is very different.
>
> So co-design is getting a shape with brain only, while our setting a robot needs both brain and muscle.
>
>
> > My understanding of controlling morphological changes is continuously searching for optimal morphologies. Can the former one be considered as a continuous loop of the latter one?
>
> Not exactly as I described above. As I mentioned above, co-design's design process never requires your muscle movement.

---

### Meta-Review · Area_Chair_E7a4 · 2023-12-11

**Metareview:**

This paper was reviewed by four experts with mixed reviews.  AC does feel that this work makes interesting contributions by introducing a new direction for the design of reconfigurable robots. The reviewers did raise some valuable concerns. The authors are encouraged to make the necessary changes and include the missing references in the final version.

**Justification For Why Not Higher Score:**

It would be great to have more real-world robot experiments (e.g. [1]).

[1] DiffuseBot: Breeding Soft Robots With Physics-Augmented Generative Diffusion Models. NeurIPS 23

**Justification For Why Not Lower Score:**

NA

---

### Decision · Program_Chairs · 2024-01-16

Accept (poster)